# Mitochondrial ATP fuels ABC transporter-mediated drug efflux in cancer chemoresistance

Emily L. Giddings[1], Devin P. Champagne[1], Meng-Han Wu[2], Joshua M. Laffin[1], Tina M. Thornton[1], Felipe Valenca-Pereira[2], Rachel Culp-Hill[3], Karen A. Fortner[1], Natalia Romero[4], James East[1,5], Phoebe Cao [2], Hugo Arias-Pulido [6], Karatatiwant S. Sidhu [7], Brian Silverstrim[1], Yoonseok Kam[4], Shana Kelley [8], Mark Pereira[8], Susan E. Bates[9], Janice Y. Bunn[10], Steven N. Fiering [6], Dwight E. Matthews [7], Robert W. Robey[11], Domink Stich[12], Angelo D'Alessandro [3✉] & Mercedes Rincon [1,2✉]

Chemotherapy remains the standard of care for most cancers worldwide, however development of chemoresistance due to the presence of the drug-effluxing ATP binding cassette (ABC) transporters remains a significant problem. The development of safe and effective means to overcome chemoresistance is critical for achieving durable remissions in many cancer patients. We have investigated the energetic demands of ABC transporters in the context of the metabolic adaptations of chemoresistant cancer cells. Here we show that ABC transporters use mitochondrial-derived ATP as a source of energy to efflux drugs out of cancer cells. We further demonstrate that the loss of methylation-controlled J protein (MCJ) (also named DnaJC15), an endogenous negative regulator of mitochondrial respiration, in chemoresistant cancer cells boosts their ability to produce ATP from mitochondria and fuel ABC transporters. We have developed MCJ mimetics that can attenuate mitochondrial respiration and safely overcome chemoresistance in vitro and in vivo. Administration of MCJ mimetics in combination with standard chemotherapeutic drugs could therefore become an alternative strategy for treatment of multiple cancers.

[1] Division of Immunobiology, Department of Medicine, Larner College of Medicine, University of Vermont, Burlington, VT, USA. [2] Department of Immunology and Microbiology, University of Colorado Denver, Anschutz Medical Campus, Aurora, CO, USA. [3] Department of Biochemistry and Molecular Genetics, University of Colorado Denver, Anschutz Medical Campus, Aurora, CO, USA. [4] Cell Analysis Division, Agilent Technologies, Lexington, MA, USA. [5] Department of Radiology, Larner College of Medicine, University of Vermont, Burlington, VT, USA. [6] Department of Microbiology and Immunology, Geisel School of Medicine, Dartmouth College, Lebanon, NH, USA. [7] Department of Chemistry, University of Vermont, Burlington, VT, USA. [8] Department of Pharmaceutical Sciences, University of Toronto, Toronto, ON, Canada. [9] Division of Hematology/Oncology, Columbia University Medical Center, New York City, NY, USA. [10] Department of Medical Biostatistics, University of Vermont, Burlington, VT, USA. [11] Laboratory of Cell Biology, National Cancer Institute, National Institutes of Health, Bethesda, MD, USA. [12] Advanced Light Microscopy Core, Neurotechnology Center, University of Colorado Denver, Anschutz Medical Campus, Aurora, CO, USA. ✉email: Angelo.Dalessandro@cuanschutz.edu; Mercedes.Rincon@cuanschutz.edu

Despite the development of novel therapies, including immunotherapies, chemotherapy prior to or following surgery remains the most commonly used systemic treatment for most cancers. However, chemotherapy treatment still faces major challenges as chemoresistance usually develops in cancer patients. In some cases, patients fail the initial chemotherapy course due to the intrinsic properties of tumor cells. Frequently, after a successful response to the initial treatment with chemotherapy, the cancer recurs following acquisition of adaptive mechanisms of chemoresistance, which can extend to families of drugs distinct from the chemotherapy used (i.e., multidrug resistance)[1]. The lack of a mechanistic understanding of the energetic implications fueling chemoresistance has hampered the development of effective strategies to overcome this major hurdle in cancer therapy. While there is growing interest in cancer cell metabolism as a novel approach to interfere with cancer growth and progression, little is known about how metabolic changes contribute to the development of chemoresistance.

Different mechanisms of chemoresistance have been identified[1], but the most common mechanism is the expression of ATP binding cassette (ABC) transporters that mediate drug efflux and decrease the intracellular accumulation of anti-cancer drugs[2,3]. ABC transporters are highly dependent on ATP since they use the energy derived from ATP hydrolysis to pump substrates out of cells[4]. Several ABC transporters have been associated with the development of multidrug resistance in cancer[1,2,5], and the most characterized are ABCB1 (also known as P-gp or MDR1), ABCG2 (BCRP), and ABCC1 (MRP1)[6–12]. A correlation between the expression of ABCB1 and ABCG2 and multidrug resistance in cancer cell lines in vitro is well established, but the correlation between the expression of these transporters and chemoresistance in cancer patients is less clear, and thereby neither can be used as predictive markers for chemoresistance[5]. Thus, other factors in addition to just their presence in chemoresistant cancer cells could contribute to the activity of these efflux pumps.

ABC transporters are energy consuming pumps that require significant amounts of ATP. Biochemical studies have indicated that up to two ATP molecules are required to efflux one molecule of substrate[13,14]. Despite this energy demand and the distinct metabolism of cancer cells, little is known about the mechanisms that regulate ABC transporter activity in the chemoresistant cancer cells other than substrate availability as well as how the activity of these transporters is regulated by the metabolic state of the cell[5]. Therefore, determining how the metabolic state of the cell affects ABC transporter activity could provide alternative pathways for the generation of novel inhibitors for these transporters to overcome chemoresistance.

Although mitochondrial respiration is the primary pathway to obtain ATP in most cells, aerobic glycolysis is also essential for cancer cells as well as other highly proliferating cell types (e.g., T cells) as a source of ATP and carbon backbones as fuel and building blocks for anabolic purposes, respectively[15–18]. While historically switching to a glycolytic metabolism at the expense of mitochondrial respiration was viewed as an approach to promote cancer progression, it is now clear that mitochondria contribute to a number of functions in cancer cells[19,20]. Several studies have shown an upregulation of the mitochondrial respiratory machinery in slow-cycling, chemoresistant melanoma cells with a quiescent phenotype generated upon drug treatment[21,22]. This has also been reported in cancer stem cells, which are also more quiescent and resistant to therapy[23]. Inhibition of the electron transport chain (ETC) with different pharmacological inhibitors in these slow-cycling resistant cells has been shown to restore drug sensitivity[22]. Interestingly, the increased response to drug treatment is thought to be due to the production of reactive oxygen species that promote the death of these quiescent cancer cells[21]. A small-molecule inhibitor of mitochondrial respiration by itself has also been shown effective in glycolysis-deficient tumor cells[24]. The regulation of mitochondrial respiration in normal-cycling chemoresistant cancer cells and the contribution of mitochondrial versus glycolytic ATP to ABC transporter-mediated chemoresistance has not been investigated.

Here we show that ABC transporters use mitochondrial ATP as the main source of energy for drug efflux in chemoresistant cancer cells. In contrast, we show that ATP from glycolysis is dispensable, therefore upregulation of mitochondrial metabolism contributes to this mechanism of chemoresistance. Methylation-controlled J protein (MCJ) encoded by the DNAJC15 gene, localizes on the inner membrane of mitochondria and acts as an endogenous brake on mitochondrial respiration by negatively regulating Complex I[25–28]. Retrospective and prospective studies have shown that loss of MCJ expression in tumors correlates with chemotherapy resistance and poor prognosis in breast and ovarian cancer patients[29,30]. Here we show that increased mitochondrial respiration in chemoresistant cancer cells is due to the lack of MCJ. Importantly, we have developed therapeutic MCJ mimetics that attenuate mitochondrial respiration and ABC transporter activity in chemoresistant cancer cells both in vitro and in vivo. Restoring MCJ function is therefore a viable therapeutic strategy to inhibit ABC transporter function and overcome chemoresistance in cancer.

## Results

**Cancer cells reprogram mitochondrial metabolism when acquiring multidrug resistance.** Increased mitochondrial respiration due to increased fluxes through the tricarboxylic acid cycle (TCA)/Krebs cycle and/or upregulation of the ETC machinery has been found in slow-cycling drug resistant cancer cells (e.g., quiescent cells). However, it remains unclear whether mitochondrial respiration is also enhanced in normal-cycling cancer cells that are multidrug resistant due to mechanisms other than a slow-cycling rate. To examine differences in mitochondrial respiration between chemosensitive and chemoresistant normal-cycling cancer cells, we used the multidrug resistant NCI/ADR-RES ovarian cancer cell line and its chemosensitive parental OVCAR-8 cell line. The difference in drug response between the two cell lines was verified using doxorubicin, a standard clinical chemotherapeutic agent and the selective agent used to obtain NCI/ADR-RES cells (Supplementary Fig. 1a). Mitochondrial respiration was examined using the Seahorse XF Cell Mito Stress test for mitochondrial oxygen consumption rates (OCR). NCI/ADR-RES cells had higher basal OCR than OVCAR-8 cells (Fig. 1a and b), indicating that overall mitochondrial respiration is increased in NCI/ADR-RES cells. In contrast to OCR, baseline extracellular acidification rate (ECAR), an indicator of the rate of glycolysis, was comparable between NCI/ADR-RES cells and OVCAR-8 cells (Fig. 1c). We also determined the OCR linked to mitochondrial ATP production by subtracting OCR after oligomycin from baseline OCR. The levels of ATP-linked respiration were significantly higher in NCI/ADR-RES cells than OVCAR-8 cells (Fig. 1b), suggesting that production of mitochondrial ATP was increased in NCI/ADR-RES cells. To further investigate the fraction of total ATP production that was derived from mitochondrial respiration we used the Seahorse XF Real-Time ATP Rate assay. The fraction of ATP production derived from mitochondria within total ATP production was increased in NCI/ADR-RES cells compared with OVCAR-8 cells (Fig. 1d). Thus, chemoresistant NCI/ADR-RES have an increased ability to undergo mitochondrial respiration and to produce ATP via OXPHOS.

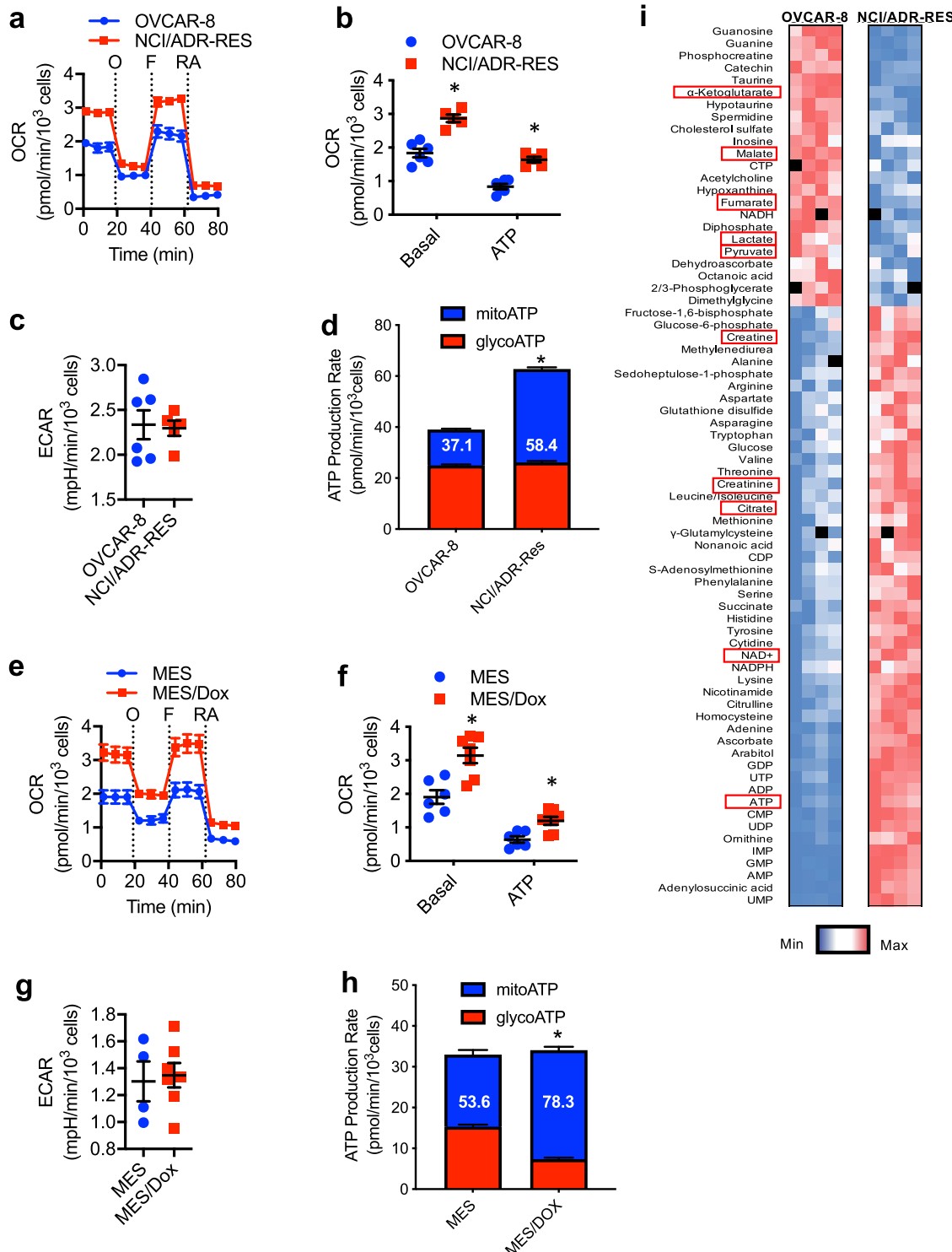

To show that increased mitochondrial respiration is also characteristic of other chemoresistant cells, we used the chemoresistant MES/Dox uterine cancer cell line and its chemosensitive parental MES-SA (MES) cell line. The difference in chemotherapy responses between the two cell lines was verified using doxorubicin (Supplementary Fig. 1b). Using the Seahorse XF Mito Stress test, MES/Dox cells were found to have higher basal OCR compared with MES cells (Fig. 1e and f), while no differences were found in baseline ECAR (Fig. 1g). In addition, ATP-linked OCR was also elevated in MES/Dox cells (Fig. 1f). Furthermore, like NCI/ADR-RES cells, the relative

contribution of mitochondrial ATP production to the total ATP production was also higher in MES/Dox cells than in their chemosensitive parental cell line (Fig. 1h). No difference in mitochondria content between chemosensitive or chemoresistant cancer cells was found based on the presence of Complex I subunits by western blot analysis (Supplementary Fig. 1c). In addition, analysis of mitochondria content by staining with Mitotracker and flow cytometry analysis further show that neither of the resistant cell lines have higher content of mitochondria relative to the sensitive cells (Supplementary Fig. 1d and e). Therefore, enhanced mitochondrial respiration

**Fig. 1 Chemoresistant cancer cells exhibit increased mitochondrial respiration and ATP production. a, b** OVCAR-8 and NCI/ADR-RES cells were analyzed by Seahorse MitoStress Assay for (**a**) oxygen consumption rates (OCR) at baseline and in response to sequential injections of oligomycin (O), FCCP (F), and rotenone with antimycin (RA). For **b** basal OCR and ATP linked (ATP) are shown (NCI/ADR-RES, $n = 5$; OVCAR-8, $n = 6$). $p = 0.0002, 0.00007$. **c** Baseline extracellular acidification rates (ECAR) of OVCAR-8 and NCI/ADR-RES cells as determined by MitoStress assay analysis. (NCI/ADR-RES, $n = 5$; OVCAR-8, $n = 6$), $p = 0.842$. **d** Mitochondrial and glycolytic ATP production rates in OVCAR-8 and NCI/ADR-RES cells as determined by Seahorse ATP Production Rate Test. Numbers in the bars represent the % of mito-ATP relative to the total. $p = 0.0001$ by two-way ANOVA. **e** OCR of MES and MES/Dox cells were determined as in (**a**). **f** Basal OCR and ATP linked (ATP) for MES cells ($n = 6$) and MES/Dox cells ($n = 7$) are shown. $p = 0.002, p = 0.005$. **g** Baseline ECAR of MES ($n = 4$) and MES/Dox ($n = 7$) cells as determined by MitoStress assay analysis, $p = 0.788$. **h** Mitochondrial and glycolytic ATP production rates in MES and MES/Dox cells as determined by Seahorse ATP Production Rate Test. $p = 0.0001$ by two-way ANOVA. **i** Relative abundances of metabolic intermediates in OVCAR-8 and NCI/ADR-RES cells as determined by mass spectrometry-based metabolomics. Within each cell type, each colum represents a independent cell preparation ($n = 4$ for each cell type). Color represents the actual value with intense blue representing the lowest value, and intense red the highest value. Specific metabolites described in the text have been framed in red. Mean ± SEM is provided for all the figures. * denotes $p < 0.05$ by unpaired $t$ test analysis for each parameter.

is a common phenotype in independently generated chemoresistant cancer cell lines from distinct origins (ovarian and uterine cancer).

Since both chemoresistant NCI/ADR-RES and MES/Dox cancer cells were derived from their parental cell lines using doxorubicin as a selective agent, we examined MCF7/Tx400 breast cancer cells that were derived from the chemosensitive MCF7 cell line using paclitaxel as a selective agent. Increased resistance to doxorubicin in MCF7/Tx400 cells was validated (Supplementary Fig. 1f). Relative to MCF7 cells, MCF7/Tx400 cells exhibited increased basal OCR (Supplementary Fig. 1g and h), but comparable baseline ECAR (Supplementary Fig. 1i). ATP-linked OCR was also higher in MCF7/Tx400 cells (Supplementary Fig. 1h). The relative contribution of mitochondrial ATP production to the total ATP production was more prominent in MCF7/Tx400 cells as well (Supplementary Fig. 1j). Thus, chemoresistant breast cancer cells also have greater capacity to produce mitochondrial ATP.

To further demonstrate a prominent selective mitochondria component in the overall cell metabolism of chemoresistant cancer cells, we performed mass spectrometry-based, high-throughput metabolic profiling[31] using chemoresistant NCI/ADR-RES cancer cells and their chemosensitive parental OVCAR-8 cells. The nonbiased metabolome analysis revealed distinct metabolic profiles for the two cancer cell lines (Fig. 1i). Metabolic sets enrichment analysis (MSEA) showed an enrichment in pathways associated with mitochondrial metabolism in NCI/ADR-RES cells relative to OVCAR-8 cells (Supplementary Fig. 2). Increased mitochondrial respiration in NCI/ADR-RES (higher OCR) is suggestive of increased fluxes through the TCA cycle, which generates the two major reduced cofactors (NADH and FADH2) that feed the ETC. Consistent with the increased mitochondrial respiration of NCI/ADR-RES described above the results from these steady-state metabolomic studies suggested that these cells could exhibit an enhanced TCA cycle activity. Thus, the levels of citrate, the first metabolite initiating TCA and derived from pyruvate, were higher in NCI/ADR-RES cells (Fig. 1i and Supplementary Fig. 3). The lower levels of pyruvate together with the lower levels of lactate in NCI/ADR-RES cells (Fig. 1i and Supplementary Fig. 3) suggested that pyruvate is used for the TCA cycle instead of generating lactate. Increased levels in high-energy phosphate compounds (e.g., ATP) as well as increase $NAD^+$ (Complex I product) (Fig. 1i and Supplementary Fig. 3) suggested an enhanced TCA cycle linked to ETC. We also found increased levels of free amino acids as well as products of amino acid catabolism and creatine/creatinine (urea cycle intermediates) potentially due to an enhanced TCA cycle in NCI/ADR-RES cells (Fig. 1i and Supplementary Fig. 4). The reduced levels of α-ketoglutarate, fumarate, and malate (Fig. 1i and Supplementary Fig. 3), downstream metabolites of citrate in the TCA cycle, could

be explained by an enhanced TCA cycle activity and a faster consumption in NCI/ADR-RES cells. To address whether there was an enhanced TCA cycle activity in NCI/ADR-RES cells we performed $^{13}C_5$-glutamine tracing experiments since glutamine feeds directly into the TCA as α-ketoglutarate through its conversion into glutamate (Supplementary Fig. 5). There were increased levels of labeled TCA cycle intermediates in NCI/ADR-RES cells (Supplementary Fig. 5).

To investigate whether this metabolic landscape could also be conserved in other chemoresistant cancer cells, we also performed the metabolome analysis in chemosensitive MES and chemoresistant MES/Dox cancer cells. Similar to NCI/ADR-RES cells, MSEA results also showed an enrichment in pathways associated with mitochondrial metabolism in MES/Dox cells relative to MES cells (Supplementary Fig. 6). Taken together, all the above studies show an enhanced mitochondrial metabolism and a predominant contribution of mitochondria to overall ATP synthesis in normal-cycling chemoresistant cancer cells.

**Mitochondrial respiration prevents drug accumulation in chemoresistant cancer cells.** NCI/ADR-RES cells are resistant to doxorubicin and other chemotherapeutic drugs because drugs fail to accumulate in the cells, and higher doses are needed to cause cell death. To investigate whether increased mitochondrial ATP production could contribute to preventing drug accumulation in NCI/ADR-RES cells, we took advantage of the intrinsic fluorescence of doxorubicin. NCI/ADR-RES cells were incubated in the presence or absence of the mitochondrial respiration inhibitor oligomycin, which blocks mitochondrial ATP synthase (Complex V). After 2 h, doxorubicin was added to the cells, and they were incubated for additional 3 h. Doxorubicin cellular accumulation was then examined by confocal microscopy. As expected, in the absence of oligomycin NCI/ADR-RES cells treated with doxorubicin contained little to no intracellular doxorubicin (Fig. 2a). Strikingly, the addition of oligomycin resulted in a pronounced accumulation of doxorubicin in the cells (Fig. 2a and b). Oligomycin had no effect on cell survival for this short period of time as determined by flow cytometry analysis (Fig. 2c). The effect of oligomycin was also observed at shorter time points of incubation with doxorubicin (Supplementary Fig. 7a). Although low doses of oligomycin known to specifically inhibit Complex V were used, to further show the contribution of mitochondrial respiration on accumulation of doxorubicin we also tested the effect of rotenone, an inhibitor of Complex I. Similar to oligomycin, rotenone treatment caused a marked accumulation of doxorubicin (Fig. 2a and b), without affecting cell survival (Fig. 2c). We also examined the effect of inhibiting mitochondrial respiration on doxorubicin accumulation by flow cytometric analysis. NCI/ADR-RES cells were treated with increasing doses of oligomycin or rotenone for

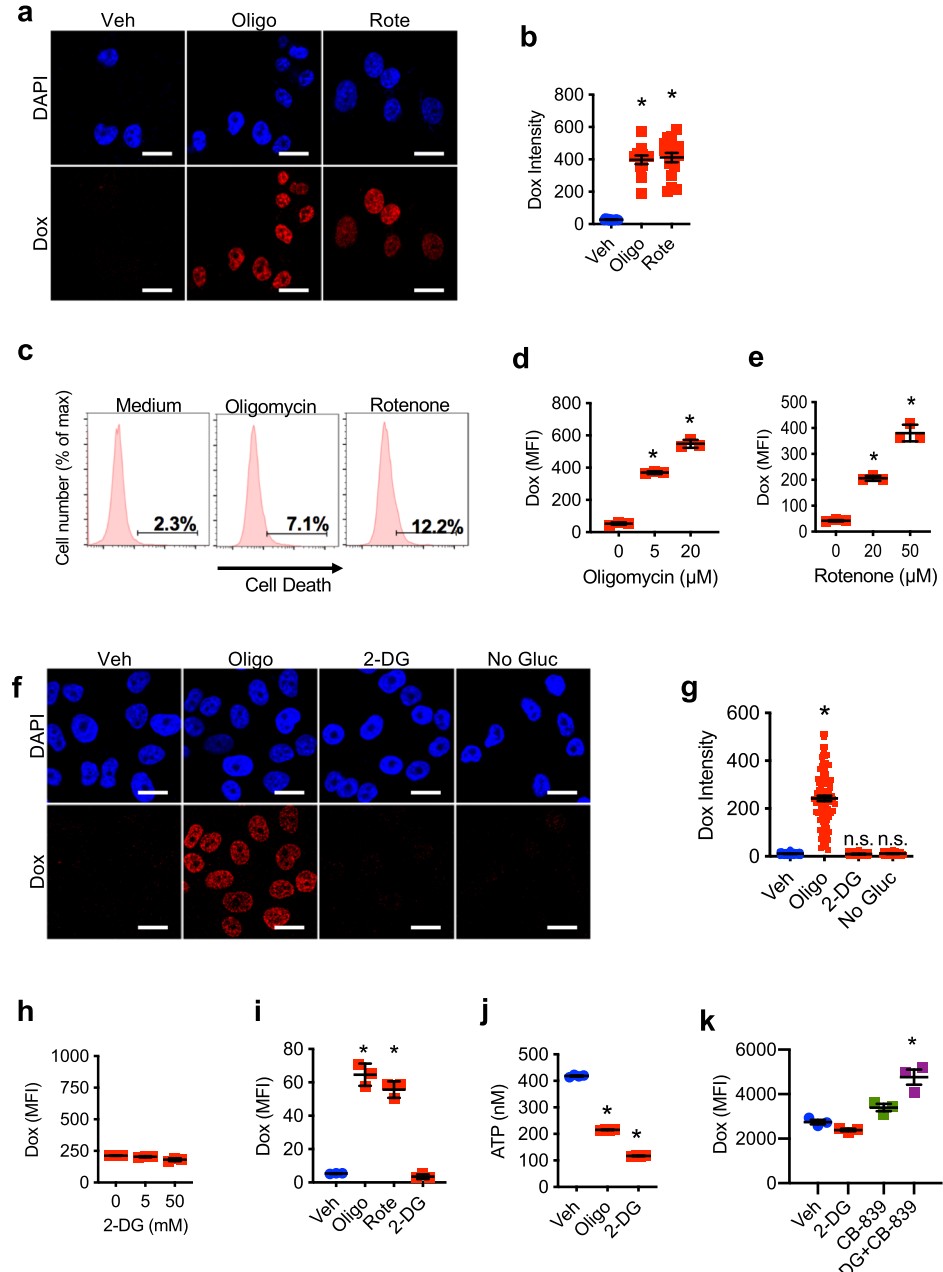

**Fig. 2 Mitochondrial respiration, but not glycolysis, is responsible for reduced drug accumulation in chemoresistant cells. a**, **b** NCI/ADR-RES cells were incubated with or without (Veh) oligomycin (Oligo, 5 μM) or rotenone (Rote, 50 μM) for 2 h followed by incubation with doxorubicin (Dox, 3 μM) for 3 h. Cells were then fixed, stained with DAPI (nuclear dye, blue), and analyzed by confocal microscopy for doxorubicin fluorescence (red). **a** Representative images. Scale bars represent 20 μm. **b** Quantification of doxorubicin intensity relative to nuclear area (vehicle $n = 12$, olligomycin $n = 12$, rotenone $n = 17$). $p = 0.0001, 0.0001$. **c** Cell death analysis using the Live/Death staining and flow cytometry analysis in cells after the treatments described in (**a**). Numbers represent the percentage of dead cells. **d**, **e** NCI/ADR-RES cells were incubated with increasing concentrations of **d** oligomycin ($p = 0.0001, 0.0001$) or **e** rotenone ($p = 0.0006, 0.0001$) as indicated for 2 h followed by incubation with doxorubicin (3 μM) as in (**a**). Cells were then fixed and analyzed for doxorubicin fluorescence by flow cytometry. Median fluorescence intensity (MFI) is shown ($n = 3$). **f**, **g** NCI/ADR-RES cells were incubated with oligomycin (5 μM) or 2-deoxyglucose (2-DG, 50 mM) for 2 h or without glucose (No Gluc) for 24 h followed by incubation with doxorubicin (3 μM) as in (**a**). Cells were then fixed, stained, and analyzed as in (**a**, **b**). **f** Representative images. Scale bars represent 20 μm. **g** Quantification of doxorubicin intensity relative to nuclear area (vehicle $n = 176$, 2-DG $n = 75$, no glucose $n = 46$, oligomycin $n = 109$). $p = 0.0001, 0.9918, 0.9999$. **h** NCI/ADR-RES cells were incubated with increasing concentrations of 2-deoxyglucose as indicated for 2 h followed by incubation with doxorubicin as (3 μM) in (**a**). Cells were then fixed and analyzed as in (**e**, **f**) ($n = 3$). $p = 0.3372, 0.0012$. **i** MES/Dox cells were incubated with oligomycin (5 μM), rotenone (50 μM), or 2-deoxyglucose (50 μM) for 2 h followed by incubation with doxorubicin as in (**a**). Cells were then fixed and analyzed as in (**d**, **e**) ($n = 3$). $p = 0.0001, 0.0001, 0.967$. **j** NCI/ADR-RES cells were incubated with oligomycin (5 μM) or 2-deoxyglucose (50 mM) for 5 h and then total ATP levels were determined by Luciferase assay ($n = 4$). ATP concentration ($10^4$ cells) is shown. $p = 0.0001, 0.0001$. **k** NCI/ADR-RES cells were incubated with 2-DG (50 mM), CB-839 (5 μM) or both for 2 h followed by incubation with doxorubicin (3 μM), and analyzed for doxorubicin fluorescence by flow cytometry. $p = 0.479, 0.1162, 0.0003$. Mean ± SEM is provided for all figures. * denotes $p < 0.05$ by one-way ANOVA and Tukey's multiple comparisons test. Results are representative of at least two experiments.

2 h followed by incubation with doxorubicin for additional 3 h. Cells were then fixed and analyzed by flow cytometry for doxorubicin fluorescence. Similar to the analysis by microscopy, both oligomycin (Fig. 2d) and rotenone (Fig. 2e) robustly increased doxorubicin accumulation in NCI/ADR-RES cells in a dose-dependent manner. Neither oligomycin or rotenone had an effect in the accumulation of doxorubicin in the parental chemosensitive OVCAR-8 cells (Supplementary Fig. 7b). In addition to production of ATP, mitochondrial respiration can be linked to production of reactive oxygen species (ROS)[32]. To test whether increase ROS, instead of mitochondrial energy, could also contribute to the effect on doxorubicin accumulation we analyzed mitochondrial ROS by staining with MitoSox in cells treated with or without oligomycin. No increase in ROS production could be detected in cells treated with oligomycin (Supplementary Fig. 7c). In addition, we also examined the effect of N-acetylcysteine (NAC), a known inhibitor of ROS, on doxorubicin uptake. Unlike oligomycin, NAC had no effect on doxorubicin accumulation (Supplementary Fig. 7d), suggesting that ROS was not responsible for the effect that inhibiting mitochondrial respiration has on doxorubicin accumulation in resistant cancer cells. Since NAD/NADH ratio is also affected by mitochondria[32] we examined whether the effect of rotenone or oligomycin on drug accumulation could be due to a major reduction on the levels of $NAD^+$, but neither of them affected the levels of $NAD^+$ during the 5 h of treatment (Supplementary Fig. 7e). Thus, together these results show that mitochondrial-derived ATP contributes to prevent doxorubicin accumulation in NCI/ADR-RES cells.

Cancer cells are typically highly dependent on glycolytic pathways for generation of ATP and carbon atom backbones to sustain anabolic demands for cell proliferation. We, therefore, examined whether glycolysis as a source of ATP was also necessary to prevent drug accumulation in chemoresistant cells. NCI/ADR-RES cells were treated with 2-deoxyglucose (2-DG), an inhibitor of glycolysis, for 2 h. As a control, we also treated cells with oligomycin as described above. Doxorubicin was added and 3 h later cells were analyzed by confocal microscopy. In contrast to oligomycin, treatment with 2-DG did not increase doxorubicin accumulation (Fig. 2f and g). As another approach to inhibit glycolysis, we incubated NCI/ADR-RES cells in medium lacking glucose for 24 h prior to addition of doxorubicin. However glucose starvation did not restore doxorubicin accumulation either (Fig. 2f and g). We performed similar experiments examining doxorubicin fluorescence by flow cytometry in NCI/ADR-RES cells treated with increasing doses of 2-DG. The results further showed that inhibition of glycolysis with a high dose of 2-DG does not restore drug accumulation in NCI/ADR-RES cells cancer cells (Fig. 2h). To investigate the contribution of mitochondrial respiration to the failure of drug accumulation occurs in other chemoresistant cells, MES/Dox cells were treated with oligomycin, rotenone, and 2-DG, followed by incubation with doxorubicin for additional 3 h and flow cytometry analysis. Similar to NCI/ADR/RES cells, treatment of MES/Dox cells with either oligomycin or rotenone caused a marked doxorubicin accumulation (Fig. 2i). In contrast, treatment with 2-DG did not restore doxorubicin accumulation (Fig. 2i). Inhibition of mitochondrial respiration with oligomycin did not affect doxorubicin accumulation in the parental MES cells (Supplementary Fig. 7f), Therefore, the regulation of drug accumulation by mitochondrial metabolism is limited to chemoresistant cancer cells, according to their enhanced mitochondrial respiration relative to the chemosensitive cancer cells.

To verify the activity of the 2-DG used for these studies we measured total ATP levels in the cells. NCI/ADR-RES cells were treated with the highest concentration of oligomycin or 2-DG used in the above assays for 5 h (same period of time used in the efflux experiments) and then total ATP levels were determined. Relative to untreated cells, ATP levels were reduced in NCI/ADR-RES cells treated with either oligomycin or 2-DG, corroborating the activity of 2-DG (Fig. 2j). Similar to oligomycin, treatment with 2-DG during the same period of time had no effect on NCI/ADR-RES cell viability (Supplementary Fig. 7g) nor caused any major energy crisis as determined by the lack of phosphorylation of AMPK (Supplementary Fig. 7h). However, after an extended period of treatment (18 h) with 2-DG cell viability/proliferation was compromised, while treatment with oligomycin for the same period of time had no effect (Supplementary Fig. 7i). Thus, mitochondrial respiration interferes with drug accumulation in chemoresistant cells but is dispensable for proliferation of the cells, while ATP derived from glycolysis is essential for proliferation/survival of these cells.

2-DG inhibits the use of glucose for glycolysis but also the use of glucose as a major carbon source for the TCA cycle through its conversion into acetyl-CoA and citrate (Supplementary Fig. 3). Therefore, the results above suggest that other carbon sources are fueled into the TCA cycle to sustain the energy requirements for chemoresistance to doxorubicin in NCI/ADR-RES cells when glucose oxidation is blocked. The results from the $^{13}C$-glutamine tracing experiments above (Supplementary Fig. 5) suggested that glutaminolysis could be enhanced during the incubation with 2-DG as compensatory mechanisms to sustain mitochondrial metabolism. As such, tracing experiments with $U$-$^{13}C_6$-glucose were performed during the incubation of NCI/ADR-RES cells with 2-DG. While 2-DG decreased glucose oxidation fluxes into the TCA cycle (Supplementary Fig. 8a), it increased the unlabeled levels of glutamine and its catabolites (glutamate), as well as unlabeled TCA cycle intermediates downstream to succinate (Supplementary Fig. 8b). Since glutamine catabolism is constrained by the activity of glutaminase, which catalyzes the rate-limiting conversion of glutamine to glutamate, treatment of NCI/ADR-RES cells with the glutaminase inhibitor CB-839 successfully ablated glutaminolysis (decreases in glutamate and increases in glutamine) in NCI/ADR-RES cells (Supplementary Fig. 8c). Inhibition of glutaminolysis also decreased the levels of TCA cycle intermediates (Supplementary Fig. 8c), suggesting that glutamine is another source fueling the TCA cycle in chemoresistant NCI/ADR-RES cells. We, therefore, investigated the effect of inhibiting both glucose and glutamine pathways on drug accumulation. NCI/ADR-RES cells were treated with 2-DG, CB-839 or the combination of both, doxorubicin accumulation was determined by flow cytometry. CB-839 alone had no effect, but the combination of 2-DG and CB-839 increased drug accumulation (Fig. 2k). In contrast, inhibiting the use of fatty acids as carbon source for the TCA cycle with etomoxir together with 2-DG did not restore doxorubicin accumulation in chemoresistant cells (Supplementary Fig. 8d). Together, these results demonstrate the selective contribution of mitochondrial respiration through glucose oxidation and glutaminolysis in preventing chemotherapeutic agent accumulation in chemoresistant cancer cells.

**Mitochondrial ATP fuels ABC transporter activity in chemoresistant cancer cells.** A major mechanism that chemoresistant cancer cells use to avoid the accumulation of chemotherapeutic drugs is the acquisition of ABC transporters that actively promote drug efflux[5]. ABC transporters utilize the energy of ATP hydrolysis to actively transport substrates against concentration gradients. While the expression of specific ABC transporters has been largely studied in chemoresistant cancer cells, including NCI/ADR-RES cells[5,33], little is known about the regulation of their activity and capacity to promote drug efflux. Since the results above show that drug accumulation in

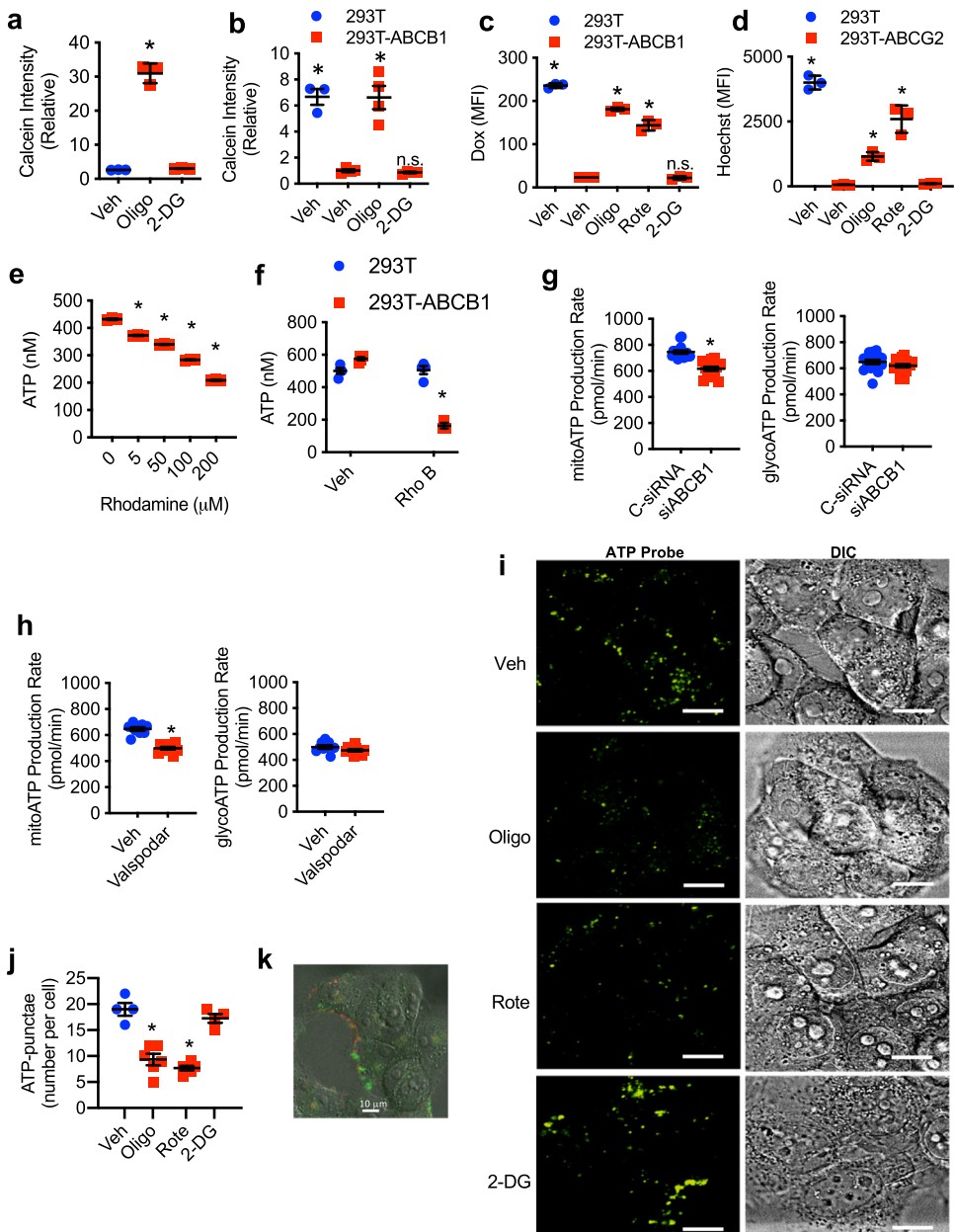

chemoresistant cancer cells is highly dependent on ATP associated with increased mitochondrial activity, we examined the contribution of these two pathways of generating energy on the activity of ABC transporters. ABCB1 is overexpressed in NCI/ADR-RES cells[33]. We, therefore, measured ABCB1 activity using a calcein retention assay. NCI/ADR-RES cells were treated with oligomycin or 2-DG for 2 h followed by incubation with calcein for 15 min, and calcein fluorescence was determined. Inhibition of mitochondrial respiration by oligomycin caused a marked reduction of ABCB1 activity as indicated by increased retention of calcein (Fig. 3a). In contrast, 2-DG treatment had no effect on ABCB1 activity (Fig. 3a). Since other mechanisms different from ABCB1 could mediate the drug resistance in NCI/ADR-RES cells, to further demonstrate the contribution of mitochondrial respiration to ABCB1 activity, we used control HEK 293T cells that do not express ABCB1 (transfected with an empty plasmid) and HEK 293T-ABCB1 cells generated by stable transfection of an ABCB1-expressing plasmid and express high levels of ABCB1 on the surface, similarly to the expression on NCI/ADR-RES cells

(Supplementary Fig. 9a). As expected, relative to control HEK 293T cells, HEK 293T-ABCB1 cells failed to accumulate calcein due to the presence of ABCB1 (Fig. 3b). However, treatment of HEK 293T-ABCB1 cells with oligomycin markedly increased calcein levels (Fig. 3b), while 2-DG treatment had no effect (Fig. 3b). Since doxorubicin is also a substrate of ABCB1, we examined doxorubicin accumulation by flow cytometry in HEK 293T-ABCB1 cells in the presence or absence of different metabolic inhibitors. As expected, almost no doxorubicin could be detected in HEK 293T-ABCB1 cells relative to control HEK 293T cells (Fig. 3c). However, treatment with oligomycin or rotenone prevented doxorubicin efflux by ABCB1 transporter, while 2-DG had no effect (Fig. 3c). To discard a potential effect that oligomycin might have on ABCB1 transporter directly instead of reducing mitochondrial ATP levels, we performed a kinetics study on NCI/ADR-RES cells expecting oligomycin to require pretreatment to lower mitochondrial ATP levels. Oligomycin or the specific inhibitor of ABCB1 valspodar were added to NCI/ADR-RES cell prior to or simultaneously to the

**Fig. 3 Mitochondrial respiration provides the energy required for ABC transporter activity. a** NCI/ADR-RES cells were treated with or without (Veh) oligomycin (Oligo, 5 μM) or 2-deoxyglucose (2-DG, 50 mM) for 2 h followed by incubation with calcein (0.25 μM) for 15 min. Cells were then washed, dissolved in DMSO, and calcein fluorescence relative to untreated cells was determined ($n = 3$). $p = 0.0001$, 0.977 by one-way ANOVA and Tukey's multiple comparisons test. **b** Calcein fluorescence in control (empty plasmid) HEK 293T cells (293T) and ABCB1-expressing HEK 293T cells (293T-ABCB1) was determined as in (**a**) (293T vehicle $n = 3$, 293T-ABCB1 vehicle $n = 4$, 293T-ABCB1 oligomycin $n = 4$, 293T-2-DG $n = 4$). $p = 0.0001$, 0.0001, 0.996 by one-way ANOVA and Tukey's multiple comparisons test. **c** 293T and 293T-ABCB1 cells were treated with oligomycin (5 μM), rotenone (50 μM), or 2-deoxyglucose (50 mM) for 2 h followed by incubation with doxorubicin (Dox, 3 μM) for 3 h. Cells were then fixed and analyzed for doxorubicin fluorescence by flow cytometry. Median fluorescence intensity (MFI) is shown ($n = 3$). $p = 0.0001$, 0.0001, 0.0001, 0.999 by one-way ANOVA and Tukey's multiple comparisons test. **d** ABCG2-expressing cells (293T-ABCG2) cells were treated with oligomycin (5 μM), rotenone (50 μM), or 2-deoxyglucose (50 mM) for 2 h followed by incubation with Hoechst 33342 (100 ng/mL) for 3 h. Control 293T cells transfected with the empty plasmid were used as positive control. Cells were then fixed and analyzed for Hoechst fluorescence by flow cytometry. Median fluorescence intensity (MFI) is shown ($n = 3$). $p = 0.0001$, 0.0066, 0.0001, 0.9998 by one-way ANOVA and Tukey's multiple comparisons test. **e** NCI/ADR-RES cells ($n = 3$) were incubated with increasing concentrations of rhodamine (Rho) B as indicated for 5 h and then total ATP levels were determined by Luciferase assay. ATP concentration ($10^4$ cells) is shown. $p = 0.0001$, 0.0001, 0.0001, 0.0001 by one-way ANOVA and Tukey's multiple comparisons test. **f** 293T and 293T-ABCB1 cells were incubated with or without rhodamine B (100 μM) for 5 h and then analyzed as in (**e**) (293T-vehicle $n = 4$, 293T rhodamine $n = 4$, 293T-ABCB1-vehiclle $n = 4$, 293T-ABCB1 rhodamine $n = 3$). $p = 0.9974$, 0.0001 by one-way ANOVA and Tukey's multiple comparisons test. **g** NCI/ADR-RES cells were transfected with siABCB1 or control siRNA (C-siRNA) and after 36 h they were incubated with calcein (1 μm) for 18 h prior to the Seahorse ATP production assay. Mitochondrial ATP production rate (siControl $n = 16$, siABCB1 $n = 14$) and glycolytic ATP production rate (siControl $n = 15$, siABCB1 $n = 16$) are shown. $p = 0.0003$ (left panel), 0.1711 (right panel) by unpaired $t$ test. **h** NCI/ADR-RES cells were incubated with calcein (1 μm) for 18 h, treated with Valspodar (4 μm) or vehicle for 3 h and assayed for mitochondrial ATP production rate (vehicle $n = 9$, Valspodar $n = 11$) and glycolytic ATP production rate (vehicle $n = 9$, Valspodar $n = 11$) using the Seahorse ATP prroduction rate assay. $p = 0.0001$ (left panel), 0.1299 (right panel) by unpaired $t$ test. **i** NCI/ADR-RES cells were incubated with oligomycin (5 μM), rotenone (50 μM), or 2-deoxyglucose (50 μM) for 2 h, stained with a fluorescent ATP probe (100 μM) for 5 min, and then analyzed by live cell confocal microscopy. ATP probe fluorescence (green) and bright light differential interference contrast (DIC) are shown. Scale bars represent 10 μm. **j** Quantification of number of ATP-puncta/cell in the images shown in (**g**) (vehicle $n = 4$, oligomycin $n = 6$, rotenone $n = 6$, 2-DG $n = 4$). $p = 0.0001$, 0.0001, 0.4856 by one-way ANOVA and Tukey's multiple comparisons test. **k** Live NCI/ADR-RES cells were incubated with anti-ABCB1 Ab (red) prior to the staining with ATP probe as described in (**i**) and analyzed by live confocal microscopy. Mean ± SEM is provided for all figures. * denotes $p < 0.05$ by unpaired $t$ test or one-way ANOVA and Tukey's multiple comparisons test. Results are representative of two experiments.

administration of doxorubicin. While inhibition of ABCB1 with valspodar promotes doxorubicin accumulation even when added simultaneously to doxorubicin, the effect of oligomycin required pretreatment prior to doxorubicin to be able to reduce mitochondrial ATP levels (Supplementary Fig. 9b). Moreover, an effect on doxorubicin uptake could be observed with a 10-fold lower dose of oligomycin (Supplementary Fig. 9c). Thus, together these results show that in these cells ABCB1 uses ATP generated by mitochondrial metabolism as a source of energy.

To investigate whether these energetic needs were also applicable to other ABC transporters and if mitochondrial respiration was required for their activity we analyzed HEK 293T cells that stably overexpress ABCG2, another ABC transporter associated with cancer chemoresistance[9], using Hoechst 33342 as a substrate for ABCG2. Relative to control HEK-293T cells that do not express ABCG2 (Supplementary Fig. 9d), the presence of ABCG2 in HEK 293T-ABCG2 cells prevented them from accumulating Hoechst (Fig. 3d). However, treatment with either oligomycin or rotenone restored the ability of HEK 293T-ABCG2 cells to accumulate Hoechst, while 2-DG had no effect (Fig. 3d), further demonstrating that, as for ABCB1, efflux activity of ABCG2 is also dependent on mitochondrial respiration. Thus, while ABC transporters can mediate drug efflux and confer resistance to chemotherapy, their activity requires ATP specifically generated by mitochondrial respiration.

To determine the actual ATP cost of ABC transporters relative to the total cellular ATP pool, we examined the impact of the high use of ABC transporters on total ATP levels. NCI/ADR-RES cells were incubated for 5 h with increasing amounts of rhodamine B, a non-chemotherapeutic substrate of ABCB1, with the goal of inducing excess ABCB1 activity mediating rhodamine efflux. Total ATP levels were then determined by Luciferase assay. Interestingly, relative to cells without substrate, rhodamine caused a substantial reduction of total ATP levels in a dose-dependent manner (Fig. 3e). Although rhodamine does not seem to have

harmful effects, to rule out potential toxic effects of high concentrations of rhodamine that may cause a reduction in the total ATP levels, we performed similar experiments using HEK 293T-ABCB1 cells and control HEK 293T cells. HEK 293T-ABCB1 cells treated with a high concentration of rhodamine also experienced a significant reduction in total ATP levels (Fig. 3f). In contrast, ATP levels in the control HEK 293T cells were not affected by the high concentration of rhodamine (Fig. 3f). Thus, as expected ABC transporters use high levels of ATP in chemoresistant cancer cells to sustain their drug efflux activity.

To further demonstrate that ABCB1 activity is fueled by mitochondrial ATP instead of glycolytic ATP we examined mitochondrial ATP and glycolytic ATP production rate using the Seahorse assay in NCI/ADR-RES cells where ABCB1 expression was silenced using siRNA. Transfection with siABCB1 reduced the cell surface expression of ABCB1 (Supplementary Fig. 10a), but it did not affect cell number (Supplementary Fig. 10b). Control and siABCB1 cells were incubated with calcein to stimulate the ABCB1 activity prior to the ATP production rate assay. Mitochondrial ATP production rate was diminished in siABCB1-transfected cells, while the glycolytic ATP production rate was not affected compared to siRNA control cells (Fig. 3g). As a complementary approach, we also examined the effect of inhibiting ABCB1 with valspodar. NCI/ADR-RES cells were incubated with calcein followed by a short treatment with valspodar or vehicle prior to the ATP production rate assay. Inhibition of ABCB1 reduced mitochondrial ATP production rate while glycolytic ATP production rate was not affected (Fig. 3h). Thus, the ATP used by ABCB1 for drug efflux in chemoresistant cells is predominantly derived from mitochondrial ATP.

Since ABC transporters have a low affinity for ATP in the absence of substrate[34], higher levels of ATP in their proximity could be needed to sustain their activity. Considering that mitochondria are highly dynamic organelles within the cell and can be found in the proximity of the plasma membrane[35], ATP

generated by mitochondrial respiration and transported from the mitochondrial matrix to the cytosol can help to locally raise the levels of ATP in the cytosol nearby mitochondria. For instance, it has been shown that mitochondria relocate to the leading edge lamellipodia where they increase ATP concentrations in migrating cancer cells[36]. Subcellular regions with high concentrations of ATP in the proximity of the ABC transporters could explain the need for mitochondrial ATP for sustaining the drug efflux activity of these transporters in cancer cells. We examined mitochondria distribution in NCI/ADR-RES cells after treatment with doxorubicin by Cox IV (mitochondria marker) immunostaining and confocal microscopy analysis. A relocation of mitochondria from mostly perinuclear in the absence of doxorubicin to broadly in the cytoplasm after treatment with doxorubicin was observed (Supplementary Fig. 10c and d). In addition, we also examine the proximity of mitochondria to ABCB1 on the cell surface by co-staining for CoxIV and ABCB1 prior to or after the treatment with doxorubicin. The fraction of cell surface ABCB1 adjacent to mitochondria increased over time after the treatment with doxorubicin (Supplementary Fig. 10e).

Using a fluorescence probe to identify ATP and ADP intracellular accumulation in live cells[25,37], we have previously shown the presence of high ATP accumulation subcellular regions that are dependent on mitochondrial respiration[25,37]. We, therefore, examined the potential presence of high ATP accumulation regions in live NCI/ADR-RES cells using this probe and confocal microscopy. ATP-rich subcellular regions were present in NCI/ADR-RES cells as determined by distinct puncta (Fig. 3i). As reported in other cells, these ATP-rich subcellular regions were dependent on mitochondrial respiration since their numbers were reduced by pretreatment of NCI/ADR-RES cells with oligomycin or rotenone for 2 h prior to analysis (Fig. 3i and j). Treatment with 2-DG did not reduce the presence of high ATP accumulation regions (Fig. 3i and j). To examine the presence of these high ATP subcellular regions in physical proximity of ABC transporters on the cytoplasmic membrane, we combined the ATP/ADP probe with immunostaining for ABCB1 in live NCI/ADR-RES cells by confocal microscopy. High ATP accumulation regions could be found in the proximity of the ABCB1 transporter at the cytoplasmic membrane (Fig. 3k and Supplementary Fig. 10f). We also used the GFP-based cytosolic ATP reporter iATPSnFR10 plasmid[38] to further show the presence of these high ATP cytoplasmic regions. iATPSnFR10 was overexpressed in NCI/ADR-RES cells by transfection and GFP fluorescence was visualized by confocal microscopy prior to or after treatment with oligomycin to reduced the glycolytic ATP. ATP puncta of GFP could also be detected in the transfected cells, but with oligomycin treatment they disappeared although the basal GFP level remained diffused in the cytosol (Supplementary Fig. 10g).

All together, the relocation of mitochondria, the proximity of mitochondria to cytoplasmic membrane ABC transporters, and the ability of mitochondria to raise local ATP levels in the cytosol could explain the need of ABC transporters for mitochondrial-derived ATP to promote drug efflux in chemoresistant cells.

**Increased mitochondrial respiration due to the loss of the Complex I-negative regulator MCJ fuels drug efflux in cancer cells.** The above results indicate that increased mitochondrial respiration contributes to chemoresistance in cancer cells by promoting drug efflux, suggesting that drug-mediated selection for genetic or epigenetic changes within cancer cells that affect mitochondrial respiration could lead to chemoresistance. The mitochondrial protein MCJ (encoded by the *DNAJC15* gene) is an endogenous negative regulator of Complex I that restricts the activity of the ETC[27]. In CD8 T cells and hepatocytes, MCJ

deficiency results in increased Complex I activity, increased mitochondrial respiration, and the presence of ATP-rich microdomains[25–27]. We and others have shown that MCJ deficiency in cancer cells causes chemoresistance using mouse models, and loss of MCJ expression in cancer cells correlates with chemoresistance in breast and ovarian cancer patients[29,30,39]. However, it remains unknown whether MCJ could act as a brake of mitochondrial respiration in cancer cells. We first examined whether MCJ acts as a negative regulator of mitochondrial respiration in chemosensitive MCF7 cells known to express high levels of MCJ[40] (Fig. 4a). We transfected MCF7 cells with an siRNA specific for MCJ (siMCJ) or a control siRNA (C-siRNA). Decreased levels of MCJ by siMCJ were verified by western blot analysis (Fig. 4a). We then examined mitochondrial respiration by measuring OCR using the Seahorse XF Cell Mito Stress assay. MCF7 cells transfected with siMCJ had higher basal OCR and ATP-linked OCR than those transfected with control siRNA (Fig. 4b and c).

To determine whether increased mitochondrial respiration caused by the loss of MCJ could also lead to the presence of ATP-rich microdomains identified in chemoresistant cells, siMCJ-transfected MCF7 cells were stained with the ATP probe and analyzed by live cell confocal microscopy. Almost no ATP puncta could be found in C-siRNA transfected cells (Fig. 4d). In contrast, MCF7 cells transfected with siMCJ had a large number of well-defined ATP-rich domains (Fig. 4d). Pretreatment of siMCJ transfected MCF7 cells with either oligomycin or rotenone for 2 h prior to analysis reduced the presence of ATP-rich domains, while treatment with 2-DG did not (Fig. 4d). Thus, increased mitochondrial respiration due to the loss of MCJ in human drug-sensitive breast cancer cells promotes the formation of mitochondrial-derived ATP-rich domains.

To demonstrate the effect of MCJ on the mitochondrial respiration of primary cancers in addition to cancer cell lines, we used mammary tumor cells from MMTV-PyMT mice (WT MMTV) and MMTV-PyMT mice crossed with MCJ-deficient mice (MCJ KO MMTV) previously generated[29]. We previously showed that mammary tumor development and tumor growth in MCJ KO MMTV mice was comparable to their development in MMTV mice, although tumors from MCJ KO MMTV mice are more resistant to doxorubicin treatment[29]. We first examined OCR in cells isolated from MMTV and MCJ KO MMTV tumors. MCJ KO tumor cells had higher basal OCR compared with wildtype MMTV tumor cells (Fig. 4e and f). Moreover, ATP-linked OCR and maximal respiratory capacities were also higher in MCJ KO cancer cells compare to wildtype cancer cells (Fig. 4f). Thus, loss of MCJ in primary cancer cells results in increased mitochondrial respiration.

Since MCJ deficiency leads to resistance to doxorubicin and other chemotherapeutic drugs, we investigated whether the lack of MCJ in MCJ KO MMTV tumor cells prevented accumulation of doxorubicin due to increased mitochondrial-derived ATP. Primary tumor cells isolated from WT MMTV and MCJ KO MMTV mice were expanded in vitro and then pretreated with oligomycin for 2 h followed by the addition of doxorubicin for 3 h. Intracellular doxorubicin accumulation was then examined by confocal microscopy. While doxorubicin was clearly present in WT MMTV tumor cells, it was almost undetectable in MCJ KO MMTV cells (Fig. 4g). However, inhibiting mitochondrial ATP production with oligomycin restored doxorubicin accumulation in MCJ KO MMTV cells (Fig. 4g). Oligomycin had little to no effect on doxorubicin accumulation in WT MMTV cells (Fig. 4g). Together these results reveal the mechanism whereby loss of MCJ as a mitochondrial regulator can cause chemoresistance in cancer cells. Loss of MCJ results in enhanced mitochondrial respiration that is necessary to fuel ABC transporter-mediated drug efflux.

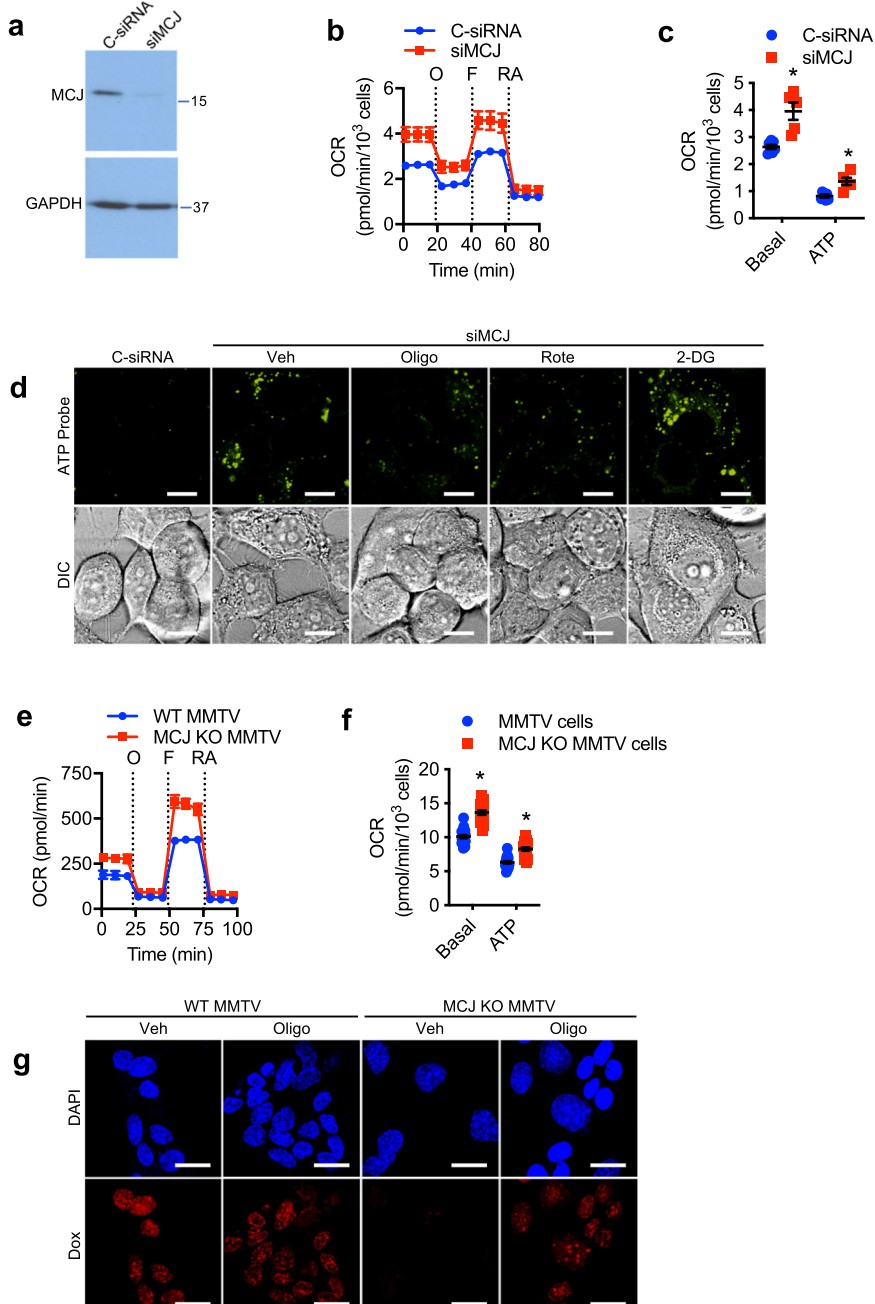

**Fig. 4 Loss of MCJ as a mitochondrial brake in cancer cells boosts mitochondrial respiration. a** MCJ (16 kDa) expression in MCF7 cells transfected with a control siRNA (C-siRNA) or an siRNA specific for MCJ (siMCJ) as determined by Western blot analysis. GAPDH (36 kDa) is shown as a loading control. **b**, **c** MCF7 cells transfected with C-siRNA or siMCJ were analyzed by Seahorse MitoStress Assay for **b** oxygen consumption rates (OCR) at baseline and in response to sequential injections of oligomycin (O), FCCP (F), and rotenone with antimycin (RA) and for **c** basal and ATP linked (ATP) relative to C-siRNA transfected cells (C-siRNA, n = 7; siMCJ, n = 5). p = 0.0008, 0.001. **d** MCF7 cells transfected with C-siRNA or siMCJ were incubated with or without (Veh) oligomycin (Oligo, 5 μM), rotenone (Rote, 50 μM), or 2-deoxyglucose (2-DG, 50 μM) for 2 h, stained with a fluorescent ATP probe (100 μM) for 5 min, and then analyzed by live cell confocal microscopy. ATP probe fluorescence (green) and bright light differential interference contrast (DIC) are shown. Scale bars represent 10 μm. **e**, **f** Cancer cells were freshly isolated from mammary tumors of WT MMTV-PyMT mice (WT MMTV) and MCJ deficient MMTV-PyMT mice (MCJ KO MMTV) and then analyzed for OCR as in (**b**, **c**) (WT MMTV, n = 22; MCJ KO MMTV, n = 20). p = 0.0001, 0.0001. **g** WT MMTV and MCJ KO MMTV cells were incubated with oligomycin (5 μM) for 2 h followed by incubation with doxorubicin (Dox, 3 μM) for 3 h. Cells were then fixed, stained with DAPI (nuclear dye, blue), and analyzed by confocal microscopy for doxorubicin fluorescence (red). Scale bars represent 20 μm. Mean ± SEM is provided for all figures. *denotes p < 0.05 by unpaired t test. Results are representative of two or three experiments.

**MCJ mimetics attenuate mitochondrial respiration in chemoresistant cells**. The results above indicate that inhibiting mitochondrial respiration could be an alternative approach to overcome cancer chemoresistance. However, rotenone and other inhibitors of the ETC are highly toxic as they potently and indiscriminately block mitochondrial respiration. Ideally, an approach that safely attenuates ETC without a full blockage could be more appropriate as a potential therapeutic. MCJ is the first identified endogenous negative regulator of Complex I and mitochondrial respiration, and it is abundantly expressed in some

of the highly metabolically active tissues (e.g., liver, heart)[27,28]. Since the absence of MCJ causes chemoresistance, we investigated whether MCJ mimetics could restore MCJ function as a brake on the ETC in chemoresistant cancer cells. The N-terminal region (35 aa) of MCJ has no significant homology to any other currently known eukaryotic protein, and it has been predicted to interact with the NDUFv1 subunit of Complex I[27]. We, therefore, designed peptide mimetics of MCJ containing the first 20 aa of the N-terminus of human MCJ (N-MCJ). Two different designs of MCJ mimetics were developed containing the same N-MCJ 20 aa but with different sequences added to mediate delivery into the cells and mitochondria (Fig. 5a and Supplementary Fig. 11a). The MITOx20 mimetic (Fig. 5a) contains the N-MCJ sequence in addition to the HIV TAT tag that is routinely used to confer cell permeability to peptides[41,42] and a mitochondrial targeting sequence (mts)[43]. For the MITOx30 mimetic, the same N-MCJ sequence was added to a previously described mitochondria-penetrating peptide (MPP)[44] that contains hydrophobic, non-canonical amino acids and provides for efficient delivery both into cells and into mitochondria (Fig. 5a). As controls for the N-MCJ mimetics we also developed two independent designs that complement each as controls: 1) control-20 (Fig. 5a) contains the N-MCJ sequence and HIV TAT tag but lacks the mts thereby N-MCJ could be delivered in the cells but not into mitochondria; 2) control-30 (Fig. 5a) contains MPP to be targeted to mitochondria but N-MCJ sequence was inversed thereby it contains the same amino acid composition but it should not be functional due to a different conformation.

We tested whether these N-MCJ mimetics could restore MCJ function in inhibiting mitochondrial respiration in cancer cells lacking MCJ using the Seahorse Cell MitoStress assay. According to the previous studies[40], the levels of MCJ expression in chemoresistant NCI/ADR-RES cells are almost undetectable relative to the chemosensitive parental OVCAR-8, cells (Supplementary Fig. 11b). We used these cells to investigate whether N-MCJ mimetics could restore mitochondrial respiration. NCI/ADR-RES cells were incubated with MITOx20 or the control peptide for 12 h prior to analysis. MITOx20-treated cells exhibited lower basal OCR relative to control-treated cells (Fig. 5b and c). MITOx20 treatment also reduced ATP-linked OCR and the maximal respiratory capacity of NCI/ADR-RES cells (Fig. 5c). To confirm that the effects of MITOx20 were due to the MCJ N-terminal component, similar experiments were performed using MITOx30 and control-30 as control peptide. NCI/ADR-RES cells treated with MITOx30 showed lower basal OCR relative to cells treated with control-30 (Fig. 5d and e). In addition, ATP-linked OCR of NCI/ADR-RES cells treated with MITOx30 were also lower compared with cells treated with Control-30 (Fig. 5e). According to the negative regulation of Complex I by MCJ, MITOx30 also reduced Complex I activity in NCI/ADR-RES cells (Supplementary Fig. 12a). We also examined whether the treatment with N-MCJ mimetics had an impact on the overall ATP levels in NCI/ADR-RES cells. Both treatment with MITOx20 and MITOx30 significantly lowered total ATP levels after 12 h of treatment compared to untreated cells (Fig. 5f). Control peptides had no effect on total ATP levels (Fig. 5g). Moreover, to show that the effect of the N-MCJ mimetics was directly on ETC activity, similar to oligomycin or rotenone, we used a modified Seahorse MitoStress assay where MITOx30 was injected instead of oligomycin (Fig. 5h). While slower and less pronounced than the effect induced by oligomycin, MITOx30 injection caused a major and prolonged reduction in baseline OCR as well as in the maximal respiratory capacity of NCI/ADR-RES cells (Fig. 5h and i). MITOx30 did not decrease ECAR (Fig. 5j), indicating that its effect in reducing OCR was not caused by a cytotoxic effect. Moreover, unlike oligomycin, MITOx30 had

no effect in cancer cells that already have endogenous MCJ, like the chemosensitive MCF7 cells, as expected (Supplementary Fig. 12b). No effect on OCR (Fig. 5i) or ECAR (Fig. 5j) by control peptides could be detected. Thus, together these results show that these N-MCJ mimetics can restore MCJ function by attenuating (but not abrogating) mitochondrial respiration in those chemoresistant cells that have lost MCJ, and would have minimal effect in normal cells expressing MCJ.

To further investigate whether the treatment with N-MCJ mimetics could also affect mitochondrial metabolism by interfering with the ETC activity, NCI/ADR-RES cells were treated with MITOx30 for 12 h and then analyzed by mass spectrometry-based metabolomics. Results showed that MITOx30 alter certain aspects of cell metabolism in the chemoresistant cells (Supplementary Fig. 13a and b). The levels of high-energy compounds (e.g., triphosphate) that were increased in NCI/ADR-RES cells relative to OVCAR-8 cells (Supplementary Fig. 3) were decreased while the levels of low energy compounds (e.g., monophosphates) were increased by MITOx30 (Supplementary Fig. 13c). Similarly, the levels of $NAD^+$ that were elevated in NCI/ADR-RES cells relative to OVCAR-8 were also decreased by MITOx30 (Supplementary Fig. 13c). MITOx30 also lowered the levels of citrate (Supplementary Fig. 13c), a primary TCA cycle intermediate that was elevated in NCI/ADR-RES cells (Supplementary Fig. 3). Overall these results indicate that N-MCJ mimetics are sufficient to attenuate some aspects of mitochondrial function that are amplified in chemoresistant cells.

**Decreased drug efflux and chemoresistance in vitro by attenuating mitochondrial respiration with N-MCJ mimetics in cancer cells.** Since ABC transporter-mediated chemoresistance is dependent upon mitochondrial respiration, and N-MCJ mimetics can attenuate mitochondrial respiration and ATP production in chemoresistant cells, we investigated whether N-MCJ mimetics could restore doxorubicin accumulation in NCI/ADR-RES cells. We chose a low concentration of MITOx20 that had no effect on the viability of NCI/ADR-RES cells by itself even after 2 d of treatment (Fig. 6a). To examine the effect of MITOx20 on drug efflux, NCI/ADR-RES cells were treated with MITOx20 for 2 h followed by incubation with doxorubicin for 3 h. Doxorubicin fluorescence was then examined by confocal microscopy. MITOx20-treated cells accumulated higher levels of doxorubicin compared with cells treated with vehicle (Fig. 6b). Similar to MITOx20, treatment with MITOx30 also restored doxorubicin accumulation in NCI/ADR-RES cells (Fig. 6c). The effect of MCJ mimetics is not caused by an effect on ABCB1 expression as determined by flow cytometry (Supplementary Fig. 14a) or a direct effect on ABCB1 (Supplementary Fig. 14b). In addition, control peptides had no effect on doxorubicin uptake (Supplementary Fig. 14c).

We further examined doxorubicin accumulation by confocal microscopy in HEK 293T-ABCB1 cells. HEK 293T cells lack MCJ expression[40]. Cells were treated with MITOx30 for 2 h followed by incubation with doxorubicin for 3 h and confocal microscopy analysis. MITOx30 treatment enhanced doxorubicin retention in HEK 293T-ABCB1 cells (Fig. 6d). Thus, by attenuating mitochondrial respiration, N-MCJ mimetics lower ABC transporter activity and drug efflux in chemoresistant cancer cells.

To determine whether the treatment with N-MCJ mimetics could overcome the resistance to doxorubicin in multidrug resistant cancer cells, NCI/ADR-RES cells were incubated with doxorubicin alone or in combination with MITOx30 or Control-30 for 3 d and then the number of viable cells was determined by Trypan blue exclusion. Neither doxorubicin, MITOx30 or Control-30 alone affected cell survival relative to untreated cells

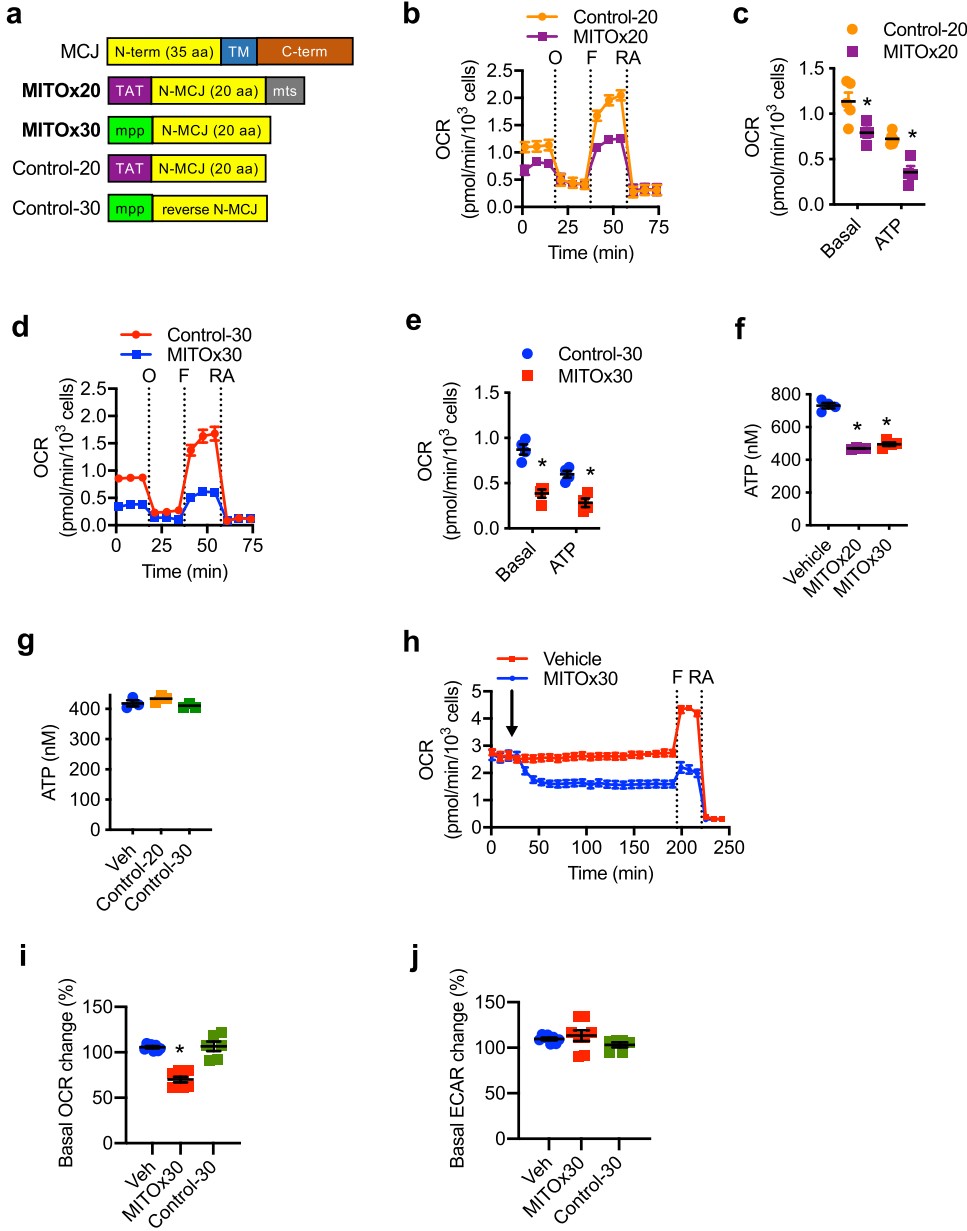

**Fig. 5 N-MCJ mimetics attenuate mitochondrial respiration in chemoresistant cancer cells. a** Diagram of N-MCJ mimetics. Top diagrams show full length MCJ (not to scale). N-term, N-terminus; TM, transmembrane domain; C-term, C-terminus; TAT, HIV TAT sequences; N-MCJ, first 20 aa of the MCJ N-terminus; mts, mitochondrial targeting sequence; mpp, mitochondrial penetrating peptide; rev, reversed aa sequence. **b, c** NCI/ADR-RES cells were treated with MITOx20 (25 μM) or Control-20 (25 μM) for 12 h and the analyzed by Seahorse MitoStress Assay for **b** oxygen consumption rates (OCR) at baseline and in response to sequential injections of oligomycin (O), FCCP (F), and rotenone with antimycin (RA) and for **c** basal (Control $n = 5$, MITOx20 $n = 4$) and ATP-linked (ATP) (Control n = 5, MITOx20 $n = 4$) relative to Control-20 treated cells. $p = 0.0243$, 0.001 by unpaired $t$ test. **d, e** NCI/ADR-RES cells were treated with MITOx30 (25 μM) or Control-30 (25 μM) for 12 h and then analyzed as in (**b, c**) ($n = 4$). $p = 0.0005$, 0.002 by unpaired $t$ test. **f** NCI/ADR-RES cells were incubated with or without (Vehicle) MITOx20 (25 μM) or MITOx30 (25 μM) for 5 h and then total ATP levels were determined by Luciferase assay ($n = 4$). $p = 0.0001$, 0.0001 by one-way ANOVA and Tukey's multiple comparisons test. **g** NCI/ADR-RES cells were incubated with or without (Vehicle) Control-20 (25 μM) or Control-30 (25 μM) for 5 h and then total ATP levels were determined by Luciferase assay ($n = 3$). $p = 0.3635$, 0.7322 by one-way ANOVA and Tukey's multiple comparisons test. **h** NCI/ADR-RES cells were analyzed for OCR at baseline and in response to sequential injections of MITOx30 (25 μM), FCCP, and rotenone with antimycin. **i** Basal OCR ($p = 0.0001$, 0.9681 by one-way ANOVA and Tukey's multiple comparisons test), **j** and ECAR ($p = 0.7301$, 0.4473 by one-way ANOVA and Tukey's multiple comparisons test) in NCI/ADR-RES cells in response to injection of MITOx30 or Control-30 as determined by MitoStress assay (vehicle $n = 8$, Control-30 $n = 6$, MITOx30 $n = 8$). Mean ± SEM is shown for all figures. * denotes $p < 0.05$ by unpaired $t$ test or one-way ANOVA and Tukey's multiple comparisons test. Results are representative of at least two experiments.

(Fig. 6e). Similarly, the combination of doxorubicin and Control-30 had no effect on cell survival (Fig. 6e). In contrast, the combination of doxorubicin with MITOx30 markedly reduced NCI/ADR-RES cell viability (Fig. 6e). Treatment with MITOx20

also enhanced the response of NCI/ADR-RES cells to doxorubicin after 2 d (Fig. 6f). N-MCJ mimetics did not have an effect on the response to doxorubicin in chemosensitive cancer cells like MCF7 or OVCAR cells (Supplementary Fig. 15a and b). To assess

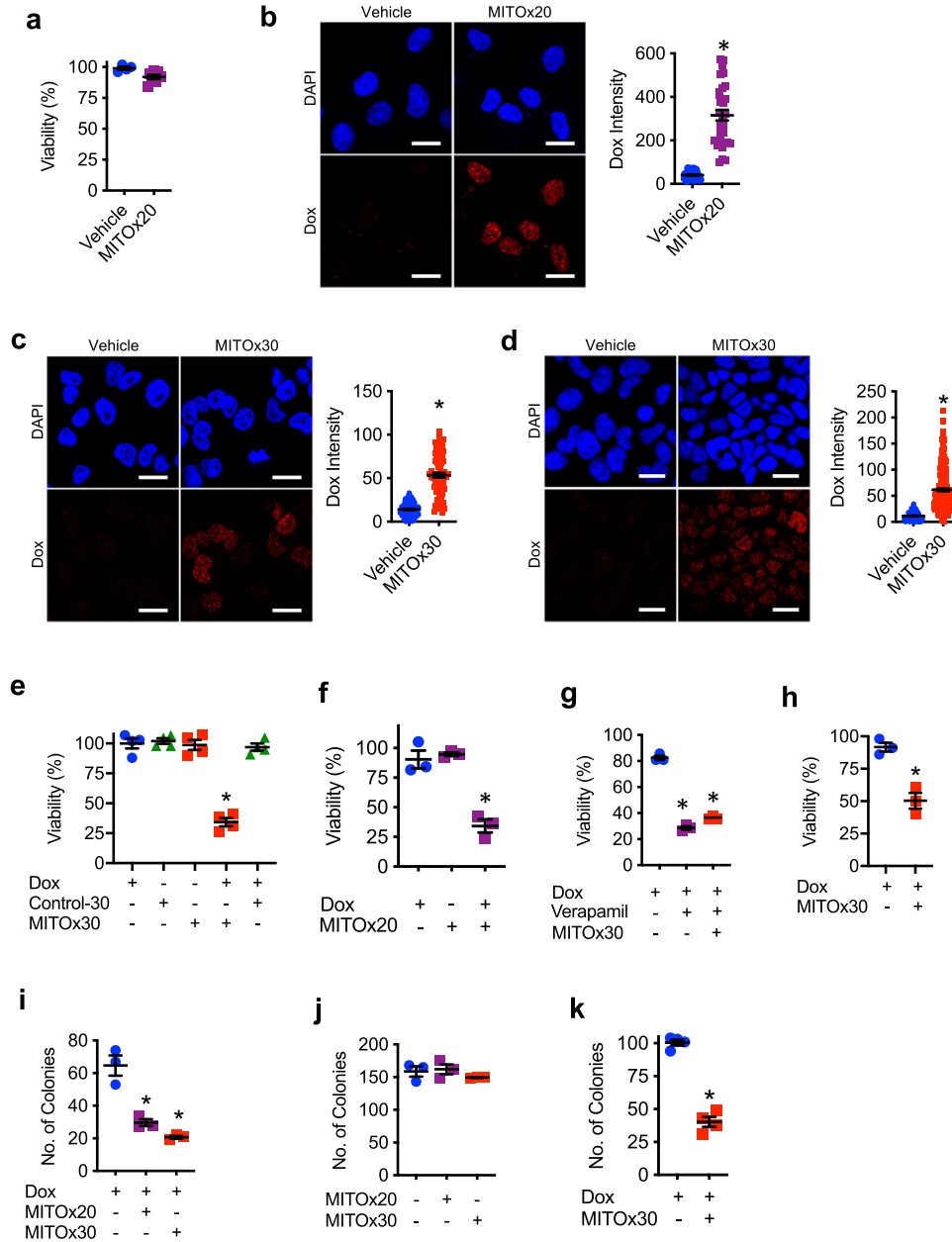

whether the effect of N-MCJ mimetics on survival of chemore-sistant cells was predominantly through their effect on ABC transporters and drug accumulation versus a systemic effect on the metabolism we tested the effect of MITOx30 in combination with verapamil, a relative broad ABC transporter inhibitor, on NCI/ADR-RES cells. The addition of MITOx30 had no effect over the effect of doxorubicin in the presence of verapamil on cell survival (Fig. 6g). We also examined the effect of N-MCJ mimetics on doxorubicin responses in the chemoresistant MES/Dox cancer cells since MCJ expression in these cells is also markedly lower than the parental chemosensitive MES cells[40] (Supplementary Fig. 11c). Consistent with results for NCI/ADR-RES cells, while doxorubicin alone had no effect, the combination of doxorubicin and MITOx30 decreased cell survival after 3 d (Fig. 6h).

To further demonstrate the effect of N-MCJ mimetics in overcoming chemoresistance in cancer cells in vitro, we performed clonogenic assays. NCI/ADR-RES cells were treated with doxorubicin alone or in combination with MITOx20 or MITOx30 for 2 d. Cells were then replated at a low density and

grown in normal culture medium. After 1 wk, cell colonies were counted. Both MITOx20 and MITOx30 in combination with doxorubicin reduced the number of colonies relative to doxorubicin alone (Fig. 6i). No effect was observed with either MITOx20 or MITOx30 alone in the absence of doxorubicin (Fig. 6j). Similarly, no effect was observed with doxorubicin and control peptides relative to doxorubicin alone (Supplementary Fig. 15c). We also examined the effect of MITOx30 in combination with doxorubicin on the clonogenicity of MES/Dox cells. Similar to NCI/ADR-RES cells, MITOx30 together with doxorubicin reduced the number of MES/Dox cell colonies relative to doxorubicin alone (Fig. 6k). Thus, together these results show the efficacy of combination therapies of N-MCJ mimetics and chemotherapy in overcoming chemoresistance of cancer cells in vitro.

**N-MCJ mimetics reverse cancer chemoresistance in vivo.** To determine whether MCJ mimetics show efficacy in reversing

**Fig. 6 N-MCJ mimetics sensitize chemoresistant cells to chemotherapy treatment. a** NCI/ADR-RES cells were treated with or without (Vehicle) MITOx20 (5 μM) for 3 d and then surviving cell counts were determined by Trypan blue exclusion. Viability relative to untreated cells is shown (Vehicle $n$ = 4, MITOx20 $n$ = 7). $p$ = 0.02 by unpaired $t$ test. **b** NCI/ADR-RES cells were incubated with MITOx20 (5 μM) for 2 h followed by incubation with doxorubicin (Dox, 3 μM) for 3 h. Cells were then fixed, stained with DAPI (nuclear dye, blue), and analyzed by confocal microscopy for doxorubicin fluorescence (red). Representative images and quantification of doxorubicin intensity relative to nuclear area are shown (Vehicle $n$ = 44, MITOx20 $n$ = 32). Scale bars represent 20 μm. $p$ = 0.0001 by unpaired $t$ test. **c** NCI/ADR-RES and **d** ABCB1-expressing HEK 293T cells were incubated with MITOx30 (5 μM) for 2 h followed by incubation with doxorubicin (3 μM) and analysis as in (**b**). Representative images and quantification of doxorubicin intensity relative to nuclear area are shown. For **c** Vehicle $n$ = 138, MITOx30 $n$ = 74. $p$ = 0.0001 by unpaired $t$ test. For **d** Vehicle $n$ = 106, MITOx30 $n$ = 284. $p$ = 0.0001 by unpaired $t$ test. Scale bars represent 20 μm. **e** NCI/ADR-RES cells were treated with doxorubicin (3 μM), Control-30 (5 μM), and/or MITOx30 (5 μM) for 3 d and then cell viability was determined as in (**a**) ($n$ = 4). $p$ = 0.9999, 0.9988, 0.993, 0.0001 by one-way ANOVA and Tukey's multiple comparisons test. **f** NCI/ADR-RES cells were treated with doxorubicin (3 μM) and/or MITOx20 (5 μM) for 2 d and then cell viability was determined as in (**a**) ($n$ = 3). $p$ = 0.8399, 0.0008, 0.0005 by one-way ANOVA and Tukey's multiple comparisons test. **g** NCI/ADR-RES cells were treated with doxorubicin (3 μM), verapamil (10 μM), and MITOx30 (5 μM) for 3 d and then cell viability was determined as in (**a**) ($n$ = 3). $p$ = 0.0001, 0.0001 by one-way ANOVA and Tukey's multiple comparisons test. **h** MES/Dox cells were treated with doxorubicin (3 μM) alone or in combination with MITOx30 (5 μM) for 3 d and then cell viability was determined as in (**a**) ($n$ = 3). $p$ = 0.0004 by unpaired $t$ test. **i** NCI/ADR-RES cells were treated with doxorubicin (3 μM) in combination with MITOx20 (5 μM) or MITOx30 (5 μM) for 2 d, replated at a low density (500 cells), grown in normal culture medium for 1 wk, and then the number of colonies formed was determined ($n$ = 4). $p$ = 0.0016, 0.0004 by one-way ANOVA and Tukey's multiple comparisons test. **j** NCI/ADR-RES cells were treated with MITOx20 (5 μM) or MITOx30 (5 μM) for 2 d and then analyzed for clonogenicity as in (**h**) ($n$ = 3). $p$ = 0.9266, 0.5766 by one-way ANOVA and Tukey's multiple comparisons test. **k** MES/Dox cells were treated with doxorubicin (3 μM) alone or in combination with MITOx30 (5 μM) for 2 d, replated at a low density (400 cells), and then analyzed for clonogenicity as in (**h**) ($n$ = 4). $p$ = 0.0001 by upaired $t$ test. Mean ± SEM is provided for all figures. * denotes $p$ < 0.05 by unpaired $t$ test or one-way ANOVA and Tukey's multiple comparisons test. Results are representative of two or three experiments.

cancer chemoresistance in vivo, we used MCJ KO MMTV mice since the mammary tumors are resistant to doxorubicin and continue to grow with the treatment[29]. Tumor cells in MCJ KO MMTV mice expressed both isoforms of the mouse ABCB1 (ABCB1A and ABCB1B) (Supplementary Fig. 16). We first tested the stability of MITOx20 and MITOx30 to serum proteases by incubating them in the presence of serum in vitro for 3 and 6 h. Spot blot analysis using a specific anti-MCJ antibody that recognizes the N-terminus showed minimal reduction even after 6 h of incubation (Fig. 7a). To rule out that treatment with N-MCJ mimetics in vivo could have an systemic effect on metabolism we followed body weight over time during the administration of MITOx20 or MITOx30 alone by subcutaneous (s.c.) administration every other day in MCJ KO or WT mice. No effect on body weight was detected (Supplementary Fig. 17a and b). Similarly, no effect of the MCJ mimetics was detected in blood glucose levels (Supplementary Fig. 17c). In addition, we found no obvious harmful effects of MITOx20 or MITOx30 in combination with doxorubicin in these mice even after the 12 d of treatment, and mice remained highly active with no signs of stress. Following H&E staining, histological analysis of the liver and heart showed no evidence of toxicity (Supplementary Fig. 18).

We, therefore, examined the effect of MCJ mimetics on the response of mammary tumors from MCJ KO MMTV mice to doxorubicin. Once tumors in MCJ KO MMTV mice reached a measurable size, mice were treated with doxorubicin alone or in combination with MITOx20, MITOx30, or Control-30. Following the doxorubicin treatment schedule used in our previous studies[29], treatments with doxorubicin (i.p.) and/or N-MCJ or control mimetics (s.c.) were given every other day for a total of 12 d after which mice treated with doxorubicin alone had to be euthanized as tumors reached the maximum approved size. Analysis of tumor volume over time showed no response of the tumor to doxorubicin alone or in combination with Control-30 ($p$ < 0.001) (Fig. 7b). In contrast, tumors in mice treated with doxorubicin and MITOx20 remained stable through the full period with no further growth ($p$ < 0.05) (Fig. 7b). In addition, tumors in mice treated with doxorubicin and MITOx30 showed a progressive reduction in size ($p$ < 0.005) (Fig. 7b). Beginning at day 6 and continuing for the duration of the experiments, the tumor volumes for animals treated with either doxorubicin and

MITOx20 or MITOx30 were significantly lower than the tumor volumes for animals treated with only doxorubicin, only MITOx30 or doxorubicin plus Control-30 ($p$ < 0.001) (Fig. 7b). After 12 days, the tumors of mice treated with doxorubicin alone or doxorubicin with Control-30 were markedly increased (Supplementary Fig 19a), while the tumors of mice treated with doxorubicin in combination with either MITOx20 or MITOx30 showed a prominent reduction in size (Supplementary Fig 19a).

To further demonstrate the potential of N-MCJ mimetics to overcome chemoresistance in human cancer, we used a xenograft model with the chemoresistant NCI/ADR-RES cells grafted into immunocompromised NSG mice commonly used for human tumor studies. Once tumors reach sufficient size, mice were treated with doxorubicin alone or in combination with MITOx20 or MITOx30. As determined by the change in tumor volume relative to the initial size, the combination of doxorubicin with either MITOx20 or MITOx30 successfully reduced the final tumor volume (Supplementary Fig. 19b). The response to doxorubicin in combination with N-MCJ mimetics was evident shortly after initiation of therapy, with a statistically significant reduction of size starting at day 6 for MITOx20 ($p$ < 0.001) and day 4 for MITOx30 ($p$ < 0.006) (Fig. 7c), while for doxorubicin alone there was a significant growth over time ($p$ < 0.001) (Fig. 7c). In contrast, Control-30 had no effect on the response of the tumors to doxorubicin (Fig. 7d). Similar to immunocompetent mice, no effect of N-MCJ mimetics on body weight was detected in NSG mice (Supplementary Fig. 17d).

Thus, together these results show that restoring MCJ function with N-MCJ mimetics increases the in vivo response to standard chemotherapy in highly resistant mouse tumors and human cancer cells that lack MCJ.

## Discussion

Chemotherapy (e.g., anthracyclines, taxanes, and cyclophosphamides) remains part of the standard of care for most types of cancers worldwide. In addition to being used as frontline therapy, chemotherapy is also used as neoadjuvant therapy to reduce tumor size prior to surgery or in combination with several of the newer biological therapies (e.g., antibodies against CTLA4, PD1,

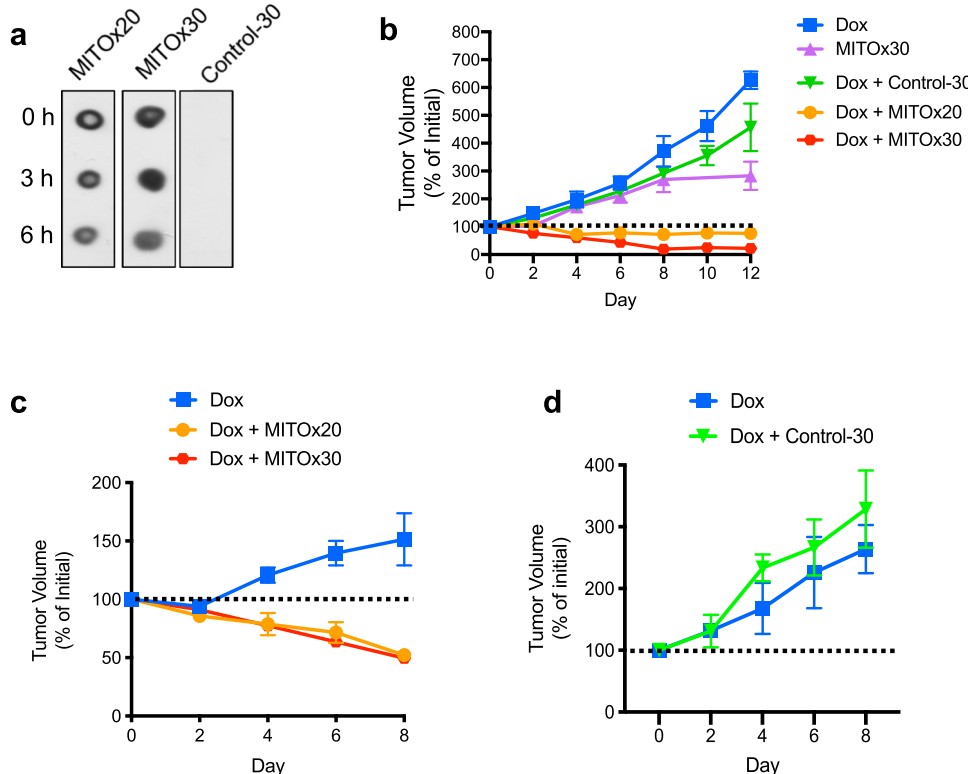

**Fig. 7 Reversal of chemotherapy resistance in mouse models of chemoresistant cancer. a** Spot blot analysis of MITOx20, MITOx30, and Control-30 prior to (0 h) or after incubation with serum (3 or 6 h) using an antibody specific for the N-terminus of MCJ. **b** MCJ-deficient MMTV-PyMT mice were treated with doxorubicin alone (Dox, $n = 5$), MITOx30 alone ($n = 4$), or doxorubicin in combination with Control-30 ($n = 4$), MITOx20 ($n = 6$), or MITOx30 ($n = 5$) every other day for 12 d. Change in tumor volume over time during treatment relative to the initial tumor size is shown. $p = 0.0001$. **c** NSG mice with NCI/ADR-RES cell xenografts were treated with doxorubicin alone or in combination with MITOx20 or MITOx30 every other day for 8 d ($n = 6$). Change in tumor volumes over time during treatment relative to the initial tumor size is shown. $p = 0.0001$. **d** NSG mice ($n = 4$ per group) with NCI/ADR-RES cell xenografts were treated with doxorubicin alone or in combination with Control-30 as in (**c**). For **b**–**d** repeated measures analysis of variance was used to examine the trajectory of tumor volume over time, and statistical significance for all post hoc tests were subject to Bonferroni correction to control for multiple comparisons. Mean ± SEM provided for all figures.

and Her2)[45]. Thus, while the development of new immunotherapy strategies (e.g., immune checkpoints and adoptive T cell therapies) is currently the main focus in cancer treatment, chemotherapy remains the most commonly used therapy for many cancers and also represents an affordable option worldwide. However, the development of chemoresistance and the resulting need for high doses of chemotherapeutic drugs with associated toxicity still represent the major limitations of this cancer treatment[1]. Unraveling the mechanisms of chemoresistance and developing novel strategies to increase chemotherapy response in cancer cells while minimizing drug toxicity is therefore still a priority[1,5]. In this study, we uncover an alternative approach to overcome chemoresistance informed by the understanding of a common mechanism that protects cancer from chemotherapy. We show that mitochondrial metabolism and mitochondrial-derived ATP are essential for the activity of the ABC transporters to promote drug efflux and chemoresistance in cancer cells. In addition, we provide an alternative target that acts as negative regulator of mitochondrial metabolism and designed compounds that restore the activity of this target and can be further developed as therapeutic agents for combination with standard chemotherapy.

ABC transporters have the ability to transport multiple substrates across the cytoplasmic membrane against the gradient by using ATP as the source of energy[46]. Some of these transporters can mediate the efflux of a number of chemotherapeutic drugs. While the correlation between the presence of these transporters

and chemoresistance in cancer cell lines is clear, the correlation between the presence of specific drug efflux transporters and poor chemotherapy response in primary tumors in cancer patients is not well established despite a number of clinical studies[5,47,48]. The presence of ABC transporters has not yet been used as biomarker for poor response to chemotherapy, and interest in these transporters in cancer has waned. Other than gene expression, little is known about the regulation of ABC transporter activity. ABC transporters are thought to have a low affinity for ATP in the absence of substrate, and up to two ATP molecules are needed for the transport of one molecule of substrate[13,14,34]. Here we reveal for the first time the unique need of ATP derived from mitochondria for the activity of ABC transporters. Interestingly, we also show that ATP derived from glycolysis is unnecessary for the activity of ABC transporters in cancer cells. Our findings could explain the paradox of why the sole presence of the ABC transporters in tumors may not correlate with poor response to chemotherapy. In line with the Warburg effect, it is well known that cancer cell metabolism is biased towards glycolysis instead of mitochondria respiration[18,49]. Thus, expression of ABC transporters in highly glycolytic cancer cells with minimal mitochondrial respiration would not be sufficient to confer chemoresistance. Here we also show that, relative to parental drug-sensitive cancer cells, derived chemoresistant cancer cells have elevated mitochondrial respiratory capacity. Thus, acquisition of ABC transporters as well as highly effective mitochondrial respiration are needed to establish chemoresistance in cancer

cells. As with other studies investigating the contribution of mitochondrial versus glycolytic ATP to specific process, a caviat in this study is that our conclusions are also based on the use of different inhibitors of the ETC or glycolysis.

Because of the potentially low affinity of ABC transporters for ATP[34], a high concentration of ATP would be needed for their activity. This requirement could be difficult to achieve only with cytoplasmic levels of ATP generated by glycolysis. It is interesting to note that, since mitochondria are dynamic organelles and can move throughout the cytosol, they can traffic to the cytoplasmic membrane where most ABC transporters localize and contribute to the generation of ATP-rich microdomains. We have previously shown the presence of ATP-rich microdomains that are dependent on mitochondrial ATP synthesis in CD8 T cells with high mitochondrial activity[25]. Here we also show the presence of these ATP-rich domains in chemoresistant cancer cell lines with high mitochondrial activity. These microdomains with high concentrations of ATP could be required to sustain the activity of ABC transporters.

MCJ has been shown to be an endogenous negative regulator of Complex I and mitochondrial respiration in a number of primary tissues including the heart, liver, and CD8 T cells[25–27]. MCJ restricts OXPHOS and mitochondrial ATP production by limiting Complex I activity[25–27]. Here we show that MCJ also acts as a negative regulator of mitochondrial respiration in cancer cells and that loss of MCJ leads to increased mitochondrial ATP production. Importantly, we show for the first time that high levels of mitochondrial-derived ATP achieved in the absence of MCJ are sufficient to fuel the activity of ABC transporters and promote drug efflux. Loss of MCJ expression correlates with poor responses to chemotherapy and poor prognoses in ovarian cancer patients[30,50]. We have shown that loss of MCJ expression causes chemoresistance in vitro in cancer cell lines and in vivo in mouse models of mammary cancer[29,40]. In addition, retrospective and prospective studies in breast cancer patients revealed that loss of MCJ in primary tumors correlates with poor responses to chemotherapy, but not with responses to hormone therapy (which is not affected by ABC transporters)[29]. It is possible that a combination of the expression of MCJ together with ABCB1 or other ABC transporters will serve as a biomarker that effectively predicts chemoresistance with high confidence. Thus, cancer cells that express drug-efflux ABC transporters, and have lost MCJ expression, will have the highest probability of chemoresistance.

A number of ABC transporters inhibitors have been developed and clinically tested, most of which interfere with substrate binding to the transporter[2,47,51]. Despite promising preclinical studies, these inhibitors failed to show efficacy in clinical trials. Toxicity due to off target effects was the main problem for first generation inhibitors of ABCB1. A lack of sufficient efficacy in combination with chemotherapy relative to chemotherapy alone was the challenge for second generation inhibitors despite significant inhibition of the three most commonly expressed ABC transporters (ABCB1, ABCG2, and ABCC1)[52,53]. The third generation of ABCB1 inhibitors showed some efficacy in clinical trials, but their high toxicity has compromised their use for cancer treatment. Thus, no ABC transporters inhibitors have been highly successful[5]. Here we show the efficacy of deliverable N-MCJ mimetics in attenuating mitochondrial respiration, reducing ABC transporter drug efflux, and increasing responses to standard chemotherapy in cancer cells in vitro. Moreover, N-MCJ mimetics showed efficacy in reducing tumor size in vivo when administered in combination with doxorubicin without increasing drug toxicity. Unlike standard inhibitors of Complex I (e.g., rotenone) or Complex V (e.g., oligomycin) that fully block the activity of these ETC complexes, MCJ is an endogenous modulator that negatively regulates Complex I by interfering with

the formation of respiratory supercomplexes but it does not fully block its activity[27]. This mechanism supports the lack of toxicity of N-MCJ mimetics. A small-molecule inhibitor of Complex I, IACS-010759, is currently in a safety and tolerability Phase I clinical trial for treatment of refractory lymphoma and advanced/metastatic solid tumors[24]. Similar to the mechanism for inhibition of Complex I by rotenone (used as a pesticide), IACS-010759 also blocks ubiquinone binding to Complex I. In contrast, MCJ binds NDUFv1 that is the Complex I subunit with NADH-dehydrogenase activity, and reduces the NADH-dehydrogenase activity[27]. We also show here that N-MCJ mimetics do not decrease mitochondrial respiration in cells expressing endogenous MCJ. Thus, N-MCJ mimetics are expected to have minimal toxicity in those tissues or cells expressing MCJ (e.g., minimal cardiotoxicity). Together, these results suggest that safely attenuating mitochondrial respiration by restoring MCJ function in combination with standard chemotherapy can be an alternative therapeutic approach in cancer patients by increasing sensitivity to lower chemotherapy doses in those cancer cells that have lost MCJ. We therefore propose that N-MCJ mimetics could be developed as an "adjuvant" chemotherapy to be administered with lower doses of standard chemotherapy.

## Methods

**Cell lines and culture conditions**. NCI/ADR-RES (formerly MCF7/Adr), MES, MES/Dox, MCF7/Tx400, were previously described[54–59]. NCI/ADR-RES cells were verified to be of ovarian origin by genotyping. MCF7 cells were purchased from American Type Culture Collection (ATCC). OVCAR-8 cells were kindly provided by Dr. Ernst Lengyel at the University of Chicago. All cancer cell lines were maintained in RPMI 1640 (Sigma R8758) containing glucose (2 mg/ml) and glutamine (0.6 mg/ml) but no pyruvate and was supplemented with 5% Fetal calf serum. HEK 293T cell lines stably transfected with an empty pcDNA plasmid (293T cells), with an ABCB1-expressing plasmid (293T-ABCB1 cells) or with an ABCG2-expressing plasmid (293T-ABCG2 cells) were previously described[57,58] and maintained in RPMI. For in vitro studies with MMTV-PyMT tumor cells, tumors were harvested from either MMTV-PyMT mice or MCJ KO MMTV-PyMT mice, and dissociated using the Tumor Dissociation kits (Miltenyi). Tumor cells were treated with doxorubicin for confocal microscopy analysis. Chemotherapeutic agents added to maintain chemoresistant phenotypes were removed at least 1 wk prior to experiments. MCF7 cell transfections were performed using siPORT NeoFX Transfection Agent (ThermoFisher Scientific) following the recommended protocol for an oligonucleotide siRNA based on the previously described MCJ targeting sequence[40]. Doxorubicin was obtained from the University of Vermont Medical Center Pharmacy. Oligomycin, rotenone, 2-deoxyglucose, paclitaxel and Hoechst 33342 were purchased from Sigma Aldrich.

**Mouse models**. All mice were maintained in the animal facility at the University of Vermont under Institutional Animal Care and Use Committee (IACUC) approved conditions. WT and MCJ deficient MMTV-PyMT were previously described[29]. NSG mice (NOD.Cg-Prkdc[scid]Il2rgtm[1Wjl]/SzJ, Jackson Laboratories) were injected with $5 - 7 \times 10^6$ NCI/ADR-RES cells (i.p.) in 150 µL PBS. For both models, mice were treated with vehicle, doxorubicin (i.p., 2 mg/kg for MMTV mice, 1 mg/kg for NSG mice), and/or and MCJ mimetics (s.c., 10 mg/kg) every other day after tumors reached 0.3–0.5 cm³. Tumor measurements were performed using a caliper to determine the tumor volumes using the formula Vol = length x width x height (above skin level). Determination of blood glucose was performed using the OneTouch Ultra glucometer and glucose strips. Animal studies were performed under the oversight of the University of Vermont IACUC and the University of Colorado IACUC guidelines.

**Extracellular flux analysis**. Oxygen consumption rates (OCR) and extracellular acidification rates (ECAR) were measured under basal conditions and in response to sequential injections of oligomycin (2 µM), FCCP (2 µM), and rotenone with antimycin A (1 µM each) using the Seahorse XF MitoStress Test Kit. Assays were performed using the medium recommended by the manufacturer (DMEM, 10 mM glucose, 2 mM glutamine, and 1 mM pyruvate, pH 7.4). OCR analyses were performed after incubation with N-MCJ mimetics for 12 h. For the direct effect of MITOx30 on OCR, MITOx30 was placed in the porter replacing oligomycin. For the experiments where two cell types are compared, number of cells in the wells after the assay was measured and OCR values were normalized to cell number. ATP production rates were determined using the Seahorse XF Real-Time ATP Rate Assay Kit following the recommended protocol. Complex I activity in NCI/ADR-RES cells was performed using cells permeabilized with XF Plasma Membrane Permeabilizer (PMP) as recommended by the manufacturer (Seahorse/Agilent)

following the previously published study[60]. Pyruvate (10 mM) was used as substrate for Complex I activity, and the MitoStress assay was performed in the presence of malate (1 mM), ADP (4 mM) and the PMP (4 nM) used. All extracellular flux analyses were performed using an XF24 or XF96 Extracellular Flux Analyzer as recommended by the manufacturer (Agilent Technologies).

**Mass spectrometry-based metabolomics**. Metabolic profile comparisons of OVCAR-8 and NCI/ADR-RES cells, or MES and MES/Dox were performed on equal numbers of cells. Cells were cultured under normal conditions, detached using trypsin-EDTA (0.05%), counted, normalized to $0.5 \times 10^6$ in each sample, and then cell pellets were snap frozen in liquid nitrogen prior to analysis. To determine the effect of N-MCJ mimetic treatment, equal numbers of NCI/ADR-RES cells were plated and allowed to grow for 2 d followed by the addition of vehicle or MITOx30 (25 μM) for 12 h. Cells were then collected and processed as above. Metabolomics and flux analyses were performed via ultra-high pressure liquid chromatography (UHPLC) coupled to high-resolution quadrupole orbitrap mass spectrometry (Vanquish—QExactive, Thermo Fisher, San Jose, CA, USA)[31,61]. Briefly, $2 \times 10^6$ cells and 20 μl of cell media were extracted in 1 mL and 980 μL of cold lysis and extraction buffer (methanol: acetonitrile: water, 5:3:2), respectively. After discarding protein pellets, 10 μL of water and methanol soluble fractions were run through a Kinetex C18 1.7 μm, $100 \times 2.1$ mm (Phenomenex) reversed phase column (Positive ion mode—phase A: water, 0.1% formic acid; B: acetonitrile, 0.1% formic acid; Negative ion mode—phase A: 1 mM $NH_4Ac$ 95:5 water: acetonitrile; phase B: 1 mM $NH_4Ac$ 95:5 acetonitrile: water) via an ultra-high performance chromatographic system (UHPLC—Vanquish, Thermo Fisher). UHPLC was coupled in line with a high-resolution quadrupole Orbitrap instrument run in both polarity modes (QExactive, Thermo Fisher) at 70,000 resolution (at 200 $m/z$) and metabolites were separated through a 5 min gradient with the phases described above and the mass spectrometer operated either in positive or negative ion mode in separate runs[62,63]. Metabolite assignment and peak integration for relative quantitation were performed through the software Maven (Princeton), against the KEGG pathway database and an in-house validated standard library (>1000 compounds, Sigma Aldrich, IROATech). Integrated peak areas were exported to Excel (Microsoft) and elaborated for statistical analysis (unpaired $t$ test) and hierarchical clustering analysis (HCA) through Prism (GraphPad Software) and GENE-E (Broad Institute), respectively.

For flux analyses, NCI/ADR-RES cells and OVCAR8 cells were grown for two days and incubated with stable isotopes substrates $^{13}C_6$-glucose (5.5 mM) (Sigma-Aldrich) or $^{13}C_5$-glutamine (650 μM) (Sigma-Aldrich) (at levels comparable to the levels in the standard culture medium) at for 5 h or 24 h, respectively. Labeling was performed in the standard culture medium. Cells were harvested and equal number of cells was used for the analyses. Data acquisition was performed by UHPLC-MS as described above for steady-state analyses[31,64]. The amount of labeled and unlabeled isotopologues for each metabolite reported in this manuscript was calculated with Maven upon correction for natural abundance of $^{13}C$[61].

Original.mzxml data were uploaded to Metabolome Xchange with the identifiers: ST001730, ST001731, ST001732 (https://www.metabolomicsworkbench.org). The mitochondria cartoon in Supplementary figures was made using the Biorender software that is licensed to the University of Colorado Mass Spectrometry Share Resources.

**Flow cytometry analysis**. Cell surface marker expression was examined by flow cytometry using directly fluorescent anti-ABCB1 (anti-CD243, ThermoFisher), and anti-ABCG2 (anti-CD338, ThermoFisher) antibodies. For detection of ABC transporter substrate accumulation, cells were pretreated with metabolic inhibitors as indicated for 2 h followed by the addition of doxorubicin (3 μM) or Hoechst 33342 (0.5 μg/mL) for 2 h. Cells were then washed, fixed in PBS supplemented with 1% paraformaldehyde, and then immediately analyzed by flow cytometry analysis. Mitochondrial mass was determining using Mitotracker dye staining (Invitrogen) and mitochondrial reactive oxygen species (ROS) was determining using MitoSox dye staining (Invitrogen) as recommended by the manufacturer. The BD Biosciences LSRII Flow Cytometer at the University of Vermont Flow Cytometry or the Cytek™ Northers LIghts Flow Cytometer at the University of Colorado School of Medicine was used. Median fluorescence intensities relative to untreated cells were determined using FlowJo Software. Histogram profiles for the specific fluorescence were obtained by gating in FSC/SSC.

**Scanning confocal microscopy analysis**. Analysis of doxorubicin pixel fluorescence intensity per nuclear area was performed using ImageJ software (version 1.51h, NIH). For analysis of subcellular ATP distribution, cells were plated in glass bottom plates (MatTek) and allowed to adhere for 2–3 d. Cells were then pretreated with metabolic inhibitors as indicated for 2 h followed by the addition of a fluorescent ATP/ADP probe[37] (100 μM) for 5 min. Cells were then washed with and immediately imaged in PBS. Analysis of ATP puncta size was performed using the Velocity Software. For analysis of doxorubicin accumulation, cells were plated on glass coverslips and allowed to grow for 3 d. Cells were then pretreated with metabolic inhibitors as indicated for 2 h followed by the addition of doxorubicin (3 μM) for 3 h. Cells were then washed, fixed in PBS supplemented with 1% paraformaldehyde, and stained with DAPI. For the ATP probe plus ABCB1 co-staining

analysis by confocal microscopy, live cells were incubated with an anti-ABCB1 mouse monoclonal Ab (C219, Biolegend) in culture for 15 min, prior to the staining with the ATP-probe. For co-staining of ABCB1 and CoxIV, performed a serial staining using the anti-ABCB1 mouse monoclonal Ab (C219, Biolegend) and the anti-CoxIV rabbit polyclonal Ab (Cell Signaling) followed by staining with DAPI as nuclear dye. Coverslips were mounted to glass microscope slides using Vectashield Antifade Mounting Medium (Vector Laboratories) for imagine imaging. Scanning confocal microscopy analyses were performed using a Zeiss LSM 510 Meta Confocal Laser Scanning microscope (Carl Zeiss Microscopy), a Nikon A1R-ER confocal microscope at the University of Vermont Microscopy Imaging Center. For analysis of ABCB1 with CoxIV co-staining and quantification of distance between mitochondria and nuclei, or ABCB1 to mitochondria, we used Zeiss LSM 780 microscope at the Advanced Light Microscopy Core part of NeuroTechnology Center at University of Colorado Anschutz Medical Campus equipped with multiple visible laser lines for scanning confocal imaging, and a pulsed tunable NIR laser (Spectra Physics MaiTai) for two-photon excitation (DAPI was imaged in two photon mode). For quantitative image analysis Fiji distribution of ImageJ was used[65–67]. Nearest neighbor analysis between the mitochondria and nuclei was implemented in an ImageJ script in the Python language Jython. The script first segments the mitochondria and nuclei channels by using Otsu thresholding which returns two binary masks in which 1's represent pixels where nuclei/mitochondria are detected. Spurious noise is removed from the masks, holes in the nuclei mask are filled and mitochondria that overlap with nuclei are excluded. The nearest neighbor search is implemented as a low-level python function. It computes the distance from each positive pixel in the mitochondria to the closest positive pixel in the nuclei mask. Distances and histograms of the distances are saved in comma-separated values files for each image file as well as the masks with their corresponding raw images. Statistics of the nearest neighbor analysis were calculated in Matlab (version R2019a). For quantification of the fraction of ABCB1 on the cell surface that was adjacent to mitochondria (CoxIV), the total ABCB1 puncta on the outline of the cells and the ABCB1 puncta adjacent to a CoxIV puncta were counted and the ratio was used for the analysis. The count was performed in a blind manner where images were randomized and assigned a number. The mean for all the areas in each of the images was used for the quantification.

**Calcein retention**. Effects of metabolic inhibitors on calcein retention were determined using the Vybrant Multidrug Resistance Assay Kit (ThermoFisher Scientific). Cells were pretreated with metabolic inhibitors as indicated for 2 h followed by the addition of calcein (0.25 μM) for 15 min. Cells were then washed, dissolved in DMSO, and then calcein absorbance was determined using an ELx800 Absorbance Microplate Reader (BioTek).

**Intracellular ATP concentration**. ATP concentration in cells ($10^4$ cells lysed in 100 ul of buffer) was determined using the ATPlite Luminescence Assay System (PerkinElmer) and a TD-20/20 Luminometer (Turner BioSystems) as recommended by the manufacturers.

**Real-time reverse transcriptase (RT)-PCR analyses**. Total RNA was isolated using an RNeasy Mini Kit (Qiagen) as recommended by the manufacturer, and cDNA was synthesized using M-MLV Reverse Transcriptase (Invitrogen)[25,29]. mRNA levels for mouse *ABCB1A* (Thermo Fisher, Mm00440761_m1) and *ABCB1B* (Thermo Fisher Mm00440736_m1) and mouse *b2-microglobulin* (Thermo Fisher Mm00437762_m1) were determined by real-time RT-PCR using the Assays–on–Demand TaqMan Gene Expression Assays (Applied Biosystems,). mRNA levels for human *MCJ* (*DnasJC15*) were determined by real-time RT-PCR using the following primer/probes:[29,40] probe, 5′-CCTTGCCAGCAGATGGG CTTACACCTAAA-3′; sense primer, 5′-CAGAAAATGAGTAGGCGAGAAGC-3′; and antisense primer, 5′-TGAC TCTCCTATGAGCTGTTCTAATC-3′. Values were normalized to $\beta_2$–macroglobulin by the comparative delta CT method.

**Western blot analyses**. Western blot analyses were performed on whole-cell lysates derived using RIPA buffer supplemented with 1 mM PMSF, 1 mM $Na_3VO_4$, and 0.5% Protease Inhibitor Cocktail (Sigma Aldrich). Lysates were separated by electrophoresis, transferred to PVDF membranes, and then analyzed using anti-actin (Santa Cruz Biotechnology), anti-NDUFA9 (MitoScience/Abcam) and anti-NDUFS3 (MitoScience/Abcam), anti-MCJ[29], anti-GAPDH (Santa Cruz), anti-AMPK (Cell Signalling), anti-phospho-AMPK (Cell Signaling) and anti-Stat1 (BD Biosciences) antibodies.

**Cell transfection**. Transfection of NCI/ADR-RES cells with human ABCB1 siRNA (118137, ThermoFisher) or control siRNA was performed using siPORT™-NeoFX™ Transfection Agent (ThermoFisher) and 30 nM or as recommended by the manufacturer. The cells were incubate at 37 °C for 2 days before performing the Seahorse ATP production rate assay as described above. Transfection of NCI/ADR-RES cells with GFP-based cytosolic ATP reporter iATPSnFR10 plasmid[38] was performed using Lipofectamine 2000 (ThermoFisher) as recommended by the manufacturer. The cells were transfected for 36 h prior to confocal microscopy imaging using the LSM Zeiss 780.

**Cell viability and clonogenicity assays**. For analysis of cell survival, cells were plated in normal culture medium and allowed to adhere for 2 d. Culture medium was then replaced with medium containing doxorubicin and/or N-MCJ mimetics, and cells were incubated for an additional 2–3 d as indicated. Cells were then washed with PBS, detached with 0.05 % trypsin-EDTA, and live cell counts were determined by Trypan blue exclusion. Viability was then calculated relative to untreated cells. Cell viability was also determined using the Live/Dead dye staining (ThermoFisher) and flow cytometry analysis as recommended by the manufacturer. For clonogenicity assays, cells were plated and allowed to adhere for 2 d followed by the addition of doxorubicin (1 μM for MES/Dox cells, 3 μM for NCI/ADR-RES cells) and/or N-MCJ mimetics (5 μM) in fresh culture medium for 2 d. Cells were then replated in normal culture medium, incubated for 1 wk, washed with PBS, fixed with PBS supplemented with 1% paraformaldehyde, stained with crystal violet (0.01% in water), and then colony counts were determined. The dose-response of the cell lines to doxorubicin was determined using the MTT assay. The cells were incubated in the presence of different concentrations of doxorubicin, (0–100 μm) for 4 days. On day 4, the remaining cells were stained with MTT (Acros Organics), the cell supernatants were aspirated, and 100 μl of DMSO were added to each well. The plates were shaken for 5 min, and the absorbances were quantified at 540 nm using a spectrophotometer (Biotek). Absorbances were normalized, and $LD_{50}$ values were calculated by nonlinear regression. Normalized data and nonlinear regression curves were plotted graphically as a percentage of viable cells.

**NAD analysis**. Intracellular NAD+ levels in NCI/ADR-RES cells ($4 \times 10^5$ cells) treated with the different inhibitors were determined using the NAD/NADH colorimetric quantitation kit (BioVision) as recommended by the manufacturer.

**N-MCJ mimetics**. The N-MCJ peptides were synthesized by GenScript. N-MCJ mimetic stability was determined by diluting stocks (1 mM in water) 1:1 in a non-identified healthy volunteer serum and incubating at ambient temperature for the indicated times. For detection, 1 μL of diluent was spotted onto nitrocellulose and allowed to air dry. Blots were then subjected to UV crosslinking for 30 s (UVP Ultraviolet Crosslinker CL-1000) and analyzed by western blot as described above. Serum was obtained from a healthy volunteer who signed a consent form that was approved by the Institutional Review Board of the University of Vermont.

**Statistical analyses**. For most of the studies in the manuscript, statistical significance was determined by unpaired two-sided $t$ test for two groups or by one-way ANOVA or two-way ANOVA with Tukey's multiple comparisons test for three or more groups using Graphpad Prism software. For metabolomics analyses, we used unpaired one-sided $t$ test. $p < 0.05$ was considered statistically significant. Figures depict mean ± standard error of mean (SEM) as described in figure legends. For metabolomics data, figures depict mean ± standard deviation (SD). For studies in Fig. 7b and c, repeated measures analysis of variance was used to examine the trajectory of tumor volume over time, and statistical significance for all post hoc tests were subject to Bonferroni correction to control for multiple comparisons.

**Reporting summary**. Further information on research design is available in the Nature Research Reporting Summary linked to this article.

## Data availability

The authors declare that all the data supporting the findings of this study are available within the paper and its Supplementary Information files. All raw data and statistical analyses for metabolomics are provided in the Supplementary tables. Original.mzxml data were uploaded to Metabolome Xchange with the identifiers: ST001730, ST001731, ST001732 (https://www.metabolomicsworkbench.org). Source data are provided with this paper.

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

## Acknowledgements

We thank Roxana del Rio-Guerra for help with the flow cytometry analysis (Flow Cytometry and Cell Sorting Facility) and Douglas Taatjes and Nicole Bouffard for help with confocal microscopy analysis (Microscopy Imaging Center) at the University of Vermont, Tinalyn Kupfer for help with the flow cytometry (Flow Cytometry facility) in the Immunology and Microbiology Department at the University of Colorado, and Dr. Matthew Jackman in the metabolic core at the University of Colorado. We thank the Advanced Light Microscopy Core part of NeuroTechnology Center at University of Colorado Anschutz Medical Campus supported in part by Rocky Mountain Neurological Disorders Core Grant Number P30 NS048154 and by Diabetes Research Center Grant Number P30 DK116073. We also thank Phani Gummadidala and Thomas Roberts (University of Vermont) for initial technical support, and Dr. James DeGregori (University of Colorado) and Michael M. Gottesman (National Cancer Institute) for critical reading. This work was supported by NIH R21 AI110016 (M.R.), NIH PO GM103496 (M.R.), R21 CA127099 (M.R.), Lake Champlain Cancer Research Organization (M.R.), Mitotherapeutix LLC (M.R.), Webb-Waring early career award from the Boettcher Foundation (A.D.), and NIH T32 AI055402 (D.P.C.).

## Author contributions

E.L.G. and D.P.C designed and performed experiments, analyzed and interpreted data, and critically wrote and reviewed the manuscript. M.-H.W., J.M.L., T.M.T., F.V.-P., R.C.-H., K.A.F., N.R., J.E., P.C., H.A-P., K.S.S., B.S., D.S., and M.P., performed experiments and analyzed and interpreted the data. J.Y.B. performed statistical analyses. Y.Y., S.K., S.E.B., D.E.W., R.W.R., and S.N.F. designed and supervised experiments. A.D. designed the research, supervised experiments, and critically wrote and reviewed the manuscript. M.R. conceived the project, designed research, supervised experiments, performed experiments, and wrote and critically reviewed the manuscript.

## Competing interests

M. Rincon and T. Thornton have a patent application related to this work. K.F. was supported by Mitotherapeutics LLC. M. Rincon is a co-founder and a member of the Scientific Advisory Board of Mitotherapeutix LLC. The other authors declare no competing interests.
