## [Peer Review File · Nature Communications]

REVIEWER COMMENTS

Reviewer #2 (Remarks to the Author):

In the revised manuscript, the authors sought to address the criticisms from the previous round of reviews. Due to the length of the authors' rebuttal, I will try to provide a broad summary with specific points that try to address the major points rather than draw out a specific point-by-point discussion. As before, I agree the experiments support the overall premise of the paper: that chemoresistant cells are disproportionately reliant on oxidative phosphorylation, thereby revealing inhibition of mitochondrial metabolism as a therapeutic target. This is supported by the data, and the authors admirably extend this to an in vivo model incorporating their longstanding work with the MCJ peptide and complex I activity.

Based on the data and justification provided in the rebuttal, I continue to think the data do not support the idea that mitochondrial ATP specifically fuels ABC transporters. This conclusion is not supported by experiments provided (even if hedged with toned-down language) and, importantly, does not need to be invoked to prove what I see as the main conclusions of the paper outlined earlier.

Following the recommendations from the reviewer, we now have performed new additional experiments suggested by the reviewer to support this conclusion. Please see below.

Some points below are as follows, though I should reiterate that these are likely immaterial if the conclusions of the manuscript are tapered back without needing to invoke the idea that ATP from mitochondria somehow specifically fuel ATP transporters.

(i) In response to the statement to reviewers: ATP is not normally kept low in the cytosol – for ATP hydrolysis to drive energetically unfavorable reactions, the ATP/ADP ratio must be kept well over 1 (it can range from ~5 to 100 in various cell types). Further evidence that ATP levels themselves are not limiting except for rare cases is given by the creatine-creatine phosphate buffering system that exists in energy demanding cells should there be local drops in ATP/ADP that affect physiology.

We apologize for the statement regarding the "low" ATP levels in our previous response. We did not refer to "low" as lower than physiological ATP levels. We referred to "low" as lower levels than those detected in the cytosol puncta in NCI/ADR-RES cells.

(ii) For those studies that do show mitochondrial motility to fuel energetic processes at particular sites (e.g. Cunniff et al. (2016) *Mol Cell Biol.* or Pekkurnaz et al. (2014) *Cell*) they refer to distal processes (lamellipodia or synapse formation down a long neuronal process) that don't seem to be relevant here. Additionally, each of these manuscripts uses a sophisticated set of quantitative experiments using time-lapse imaging and microfluidic cell growth chambers (in Pekkurnaz et al.) to definitely show this. The work presented here does not have the same level of detail to prove this point, and is rather a fixed image showing some mitochondria can localize near ABCB transporters on the plasma membrane. It lacks quantitation relative to distribution across

the cytosol and time-lapse imaging showing migration occurs in response to activity like the previously cited work.

We have now designed and performed the additional experiments suggested by the reviewer to obtain a distribution of mitochondrial localization in the context of ABCB1 function. As the reviewer described, in our previous submission we showed the presence of some mitochondria in the proximity of ABCB1 in NCI/ADR-RES cells in medium. We have now examined mitochondria distribution in these cells before and after treatment with doxorubicin to stimulate ABCB1 activity. While most mitochondria display a perinuclear distribution in NCI/ADR-RES cells without ABCB1 substrate (correlating with our previous results), after treatment with doxorubicin we can see a re-distribution of mitochondria through the cytosol and towards the cytoplasmic membrane. We now provide representative images (**new Supplementary Fig. S10c**). In addition, with the help of experts at the University of Colorado microscopy core, we have developed specific analyses for quantification of mitochondria distribution (as requested by the reviewer) and now provide these analyses showing statistically significant redistribution of the mitochondria away from the nucleus after doxorubicin treatment (**new Supplementary Fig. S10d**). We have also performed analyses to investigate the proximity of mitochondria to ABCB1 on the cell surface in response to doxorubicin. We now show an increased number of mitochondria in the proximity of ABCB1 after doxorubicin treatment (**new Supplementary Fig. S10c and S10e**). In addition to representative images, we also provide quantification (**new Supplementary Fig. S10e**). We thank the reviewer for requesting these new analyses that we had not previously considered. The results further support our previous conclusions and improve the quality of our studies.

(iii) Intracellular calcium dynamics and its unique properties (stored in the ER, gated by SERCA, taken up by mitochondria, etc.) is a bit of a non-sequitur, apples-to-oranges comparison to ATP since they have dramatically different roles in cell biology.

This comment was made in response to our previous response. We kindly respect the reviewer's opinion. Calcium dynamics were not discussed in the previous revised version of the manuscript, and nor in this current version. Thus, no additional changes have been made in this revision manuscript.

(iv) It should be noted that there is an opposing school of thought. Due to proximity, the lack of compartmentation, and faster turnover, many think it is glycolysis that mainly fuels plasma membrane ATPases such as the Na⁺/K⁺ ATPase (see James JH (1996) JCI). As such, I believe demonstrating the converse requires far more experiments along the lines of what was discussed in (ii).

We agree with the reviewer that some studies (such as James et al 1996) have suggested that plasma membrane Na⁺/K⁺ ATPase is fueled by glycolysis. However, there are other older studies (Harris SI et al JBC 1981) indicating that this ATPase is fueled by mitochondrial respiration, in that inhibition of Na⁺/K⁺ ATPase results in a decrease of mitochondrial respiratory rate, while stimulating Na⁺/K⁺ ATPase activity results in increased respiration, as the reviewer described in point (v). In addition, a more recent study (Fernandez-Moncada et al, Biochem J. 2014) concludes that the "nature of the functional coupling between

the Na/K ATPase pump and the glycolytic machinery is not energetic and that the pump is mainly fueled by mitochondria ATP". Thus, the dependency of the activity of other cell membrane pumps on glycolysis is not well established, and there is evidence that other pumps in addition to ABC transporters are also fueled by mitochondrial ATP. If the reviewer and/or editor believes we should include a paragraph in the Discussion to discuss the use of mitochondrial ATP or glycolytic ATP by other transporters, we will be happy to include it.

Additional comments in response to the rebuttal are as follows:

(v) Genetic silencing of the ABC transporters showing P-gp can indirectly show that activity is exclusively fueled by mitochondria indirectly with a Seahorse assay. Upon genetic silencing, only rates of oxidative phosphorylation should decrease, whereas glycolysis should be unchanged. A similar approach has been used many times to show how much mitochondrial ATP is used to fuel different processes (see Birket et al. J Cell Science, Fig. 5).

We apologize for our misunderstanding of the reviewer point in the previous revision. Now, we understand what the reviewer is referring to. We appreciate the reviewer providing the reference for clarification. This is indeed an interesting question. The processes assayed in the Birket et al study are essential and importantly constitutive processes in the cells (e.g. protein synthesis, transcription etc). However, in our case, P-gp (ABCB1) activity increases only when there are substrates present (in order to detoxify cells); therefore, silencing ABCB1 should not have a significant impact on the energetics of the NCI/ADR-RES cells in the absence of substrate. Thus, to address the question from the reviewer and further demonstrate the use of oxidative phosphorylation by ABC transporters, we have now used the Seahorse ATP production rate assay looking at mitochondrial-ATP production and glycolytic ATP production rate (as the reviewer suggested) in the NCI/ADR-RES cells in the presence of the ABCB1 substrate calcein. Following the reviewer's recommendations, we have silenced ABCB1 (**new Supplementary Fig. S10a and S10b**) and examined ATP production rate in the presence of calcein. The results show a decrease in mitochondrial ATP production rate, but no effect on glycolysis-ATP rate, in the ABCB1 knockdown cells (**new Fig. 3g**). In addition, to complement these studies we have also performed similar experiments using Valspodar, an inhibitor of ABCB1 in the presence of calcein. We found that in the presence of Valspodar, there is a reduction in mitochondrial ATP production rate, but there is no effect on glycolysis-ATP rate (**new Fig. 3h**). These new results clearly support the conclusion that mitochondrial respiration (but not glycolysis) meets most of the ATP demand from ABCB1 activity. We thank the reviewer for the suggestion of these experiments.

(vi) While it is agreed that glycolysis can increase in cells in response to oligomycin, it seems unlikely that it can compensate for the large change in ATP production based on Figure R1 – respiration drops substantially and ECAR increases less than 50%. So how can the authors discriminate between a compromised cell not able to make enough ATP via glycolysis (but still 'viable' by dye exclusion) vs. a healthy cell that specifically channels mitochondrial ATP to fuel the ABC transporters? Perhaps the authors could use the ATP calculations presented previously +/- oligomycin if desired to show the cells are not energetically compromised in response to mitochondrial inhibitors?

We apologize if our response is not what the reviewer was asking for but the point was not clear to us. As the reviewer described, we have shown that treatment with oligomycin does not affect cell viability using the Live/Death staining commonly used to examine cell death without specifying the type of death (provided in the previous submission). We believe it is fair to say that cells treated with oligomycin remain alive. Regarding the ATP calculations with or without oligomycin, our data in Fig. 2j indicate that treatment with oligomycin for 5 h reduces the levels of total ATP by about 50%, while 2-DG treatment for the same period of time reduces the total ATP levels by 75%. Thus, these data further support that NCI/ADR-RES cancer cells treated with oligomycin are less energetically compromised than when they are treated with 2-DG. Nevertheless, reduction of mitochondrial ATP with oligomycin (but not 2-DG) impairs ABC transporter activity and reverses drug accumulation. We hope these data answer the reviewer's question.

(vii) The use of the fluorescent dye for ATP in this context needs far more controls to be as meaningful as claimed. The authors use the dye at a 10-fold higher concentration (100 μ M vs. 10 μ M) than the manuscript cited in the methods (ref. 37). Additionally, this probe clearly measures ADP as well according to the citation, and how these concentrations of the fluorophore discriminate ATP from ADP and respond to changes in cytoplasmic pH, NAD/NADH upon use of mitochondrial inhibitors is not demonstrated. More sensitive probes exist for ATP (Lobas MA et al (2019) Nature Communications) and imaging of the ATP/ADP ratio is possible with Perceval:HR (Tantama et al. (2013) Nature Communications). Of note, ETC inhibitors will change the NAD/NADH ratio in a compartmentalized fashion (the mitochondrial matrix will reduce but the cytoplasm will oxidize) so this may affect how the fluorophore behaves even if no change is observed in the aggregate whole cell.

We kindly disagree with the reviewers since in the manuscript by Rao et al that we refer in our manuscript also used 100 μ M of the same ATP/ADP probe (Fig. 4 legend in their manuscript). We indeed followed their protocol since the authors of the referred manuscript were our collaborators in another study we published with this probe (Champagne et al. Immunity 2016). In this previous manuscript we also show similar accumulations of ATP probe in the cytosol of CD8 cells with high mitochondrial respiration and the disappearance by the treatment with oligomycin. The paper by Rao et al and our Immunity paper represent two published reports validating the ability of this probe to detect ATP in cells. We agree with the reviewer that this probe is for ATP and ADP and we acknowledge this in our manuscript, as recommended in previous revisions of this manuscript. The advantage of this probe is that manipulation of the cells is not required, such as transfection to overexpress a reporter.

Nevertheless, following the reviewer's suggestion, we have obtained an ATP-reporter DNA plasmid from the Lobas et al paper that is commercially available. Most of the studies in the manuscript were performed with a cell surface-expressed reporter but the authors have also generated a cytosolic GFP construct where the fluorescence is regulated by ATP binding. The system is somewhat more complicated since there is always fluorescence of GFP independently of ATP. We did transfection with this reporter plasmid in NCI/ADR-RES cells and after 24 h of transfection we treated cells with medium or oligomycin. Similar to our results with the fluorescent probe, we can detect foci of ATP accumulation in the cytosol, which are reduced when cells are treated with oligomycin (**new Supplementary Fig. S10g**).

(viii) Lastly and as suggested previously, the use of mitochondrial inhibitors (and high concentrations) is rife with caveats detailed in the previous review. I think it would be essential to address this in the discussion.

Following the suggestion from the reviewer we have now included this caveat in the discussion (page #28).

Reviewer #3 (Remarks to the Author):

The authors addressed satisfactorily all my previous concerns. The manuscript has significantly improved. Congratulations to the authors for the very nice piece of work.

We are pleased we addressed all the previous concerns and the reviewer considers our study appropriate for publication. We thank the reviewer.

REVIEWER COMMENTS

Reviewer #2 (Remarks to the Author):

The authors have satisfactorily addressed each of the issues presented and no additional comments are warranted or appropriate. I thank them for their efforts and recommend the manuscript as suitable for publication.